# iVGR: Internalizing Visually Grounded Reasoning for MLLMs with Reinforcement Learning

**Chang-Bin Zhang** [1]  **Yujie Zhong** [2]  **Qiang Zhang** [3]  **Kai Han** [1]

## Abstract

While visually grounded Chain-of-Thought (CoT) has emerged as a promising paradigm to enhance fine-grained perception in multimodal large language models (MLLMs), its efficacy during the inference phase remains underexplored. In this work, we empirically find that mandating explicit object boxes in visually grounded CoT during inference often degrades performance compared to standard textual CoT, which reasons without explicit visual grounding. We hypothesize that the visual localization capability can be internalized into the textual CoT and that the mandatory explicit grounding introduces unnecessary interference with the model's primary objective of answer prediction. To address this problem, we propose Internalizing Visually Grounded Reasoning (**iVGR**), a novel reinforcement learning framework that transfers localization capabilities into the textual reasoning process. We employ a dual-stream training strategy, where a textual stream is aligned with a high-quality visually grounded stream via a proposed consistency reward, enabling the model to localize accurately without explicit grounding during inference. Extensive experiments demonstrate that our method significantly outperforms existing baselines on fine-grained benchmarks, while maintaining the flexibility to support tool-assisted inference workflows. Project page: https://visual-ai.github.io/ivgr/

## 1. Introduction

Multimodal large language models (MLLMs) (Liu et al., 2023; Team et al., 2025; Yang et al., 2025) have achieved

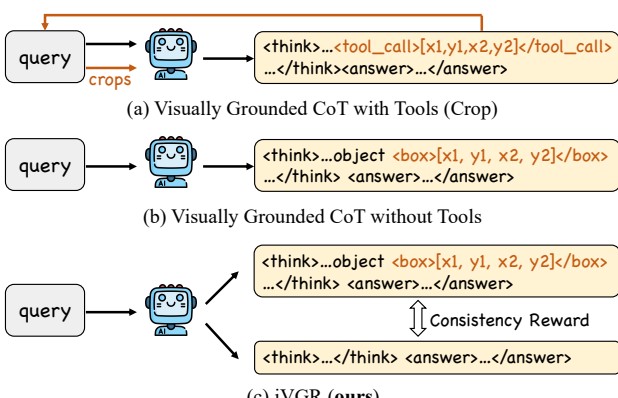

Figure 1. **Paradigms of visually grounded reasoning.** (a) **Tool-based approaches** rely on dynamically invoking crop tools to acquire fine-grained visual details. (b) **Explicit grounding approaches** mandate the generation of bounding boxes interleaved within the CoT, eliminating external tools. (c) **iVGR (ours)** introduces a dual-stream training strategy. By utilizing a consistency reward to align the textual stream with a high-quality grounded stream, we explicitly internalize localization capabilities into the textual reasoning process.

remarkable progress in recent years. While post-training techniques, such as those based on reinforcement learning (RL) (Shao et al., 2024; Yu et al., 2025b; Zheng et al., 2025a), significantly enhance their general reasoning capabilities, these models still encounter substantial challenges when processing high-resolution images or complex visual scenes (Zheng et al., 2025b; Wang et al., 2025e). In such fine-grained scenarios, standard supervision with textual Chain-of-Thought (CoT) often fails to guide MLLMs to localize and focus on critical visual details accurately. To address this, recent studies introduce visually grounded CoT (Wang et al., 2025d; Zheng et al., 2025b), which explicitly anchors the reasoning process using dynamic crops (see Figure 1(a)) or bounding boxes (see Figure 1(b)), aiming to enhance precise visual perception. Specifically, crop-based methods (e.g., DeepEyes (Zheng et al., 2025b)), as illustrated in Figure 1(a), require the MLLM to invoke an external tool to crop the target region, subsequently integrating the cropped view with the original context for downstream reasoning. Conversely, box-based methods (e.g., TreeVGR (Wang et al., 2025a)), as shown in Figure 1(b),

---

[1]Visual AI Lab, The University of Hong Kong [2]Independent Researcher [3]University of Science and Technology of China. Correspondence to: Kai Han <kaihanx@hku.hk>.

*Proceedings of the 43rd International Conference on Machine Learning*, Seoul, South Korea. PMLR 306, 2026. Copyright 2026 by the author(s).

*Table 1.* **Performance comparison between visually grounded CoT and textual CoT.** We evaluate off-the-shelf models, Deep-Eyes (Zheng et al., 2025b) and TreeVGR (Wang et al., 2025a), across two reasoning modes: standard textual CoT ('T') and visually grounded CoT ('G').

| Benchmarks | Qwen2.5-VL-7B | DeepEyes-7B | | TreeVGR-7B | |
|---|---|---|---|---|---|
| CoT | T | G | T | G | T |
| V* | 78.5 | **82.7** | 81.7 | 83.8 | **84.3** |
| HR4K | 69.0 | **75.1** | 74.9 | **77.1** | 76.9 |
| HR8K | 65.1 | 72.6 | **73.1** | 73.1 | **74.7** |
| MME-RW-Lite | 44.5 | 53.2 | **53.5** | 54.9 | 54.7 |
| POPE | 86.3 | 87.7 | **89.2** | 87.3 | **88.4** |
| RealWorldQA | 68.1 | 69.4 | **69.7** | 67.3 | 69.5 |
| CV-Bench-2D | 75.7 | 75.0 | **77.9** | 76.6 | 77.7 |
| CV-Bench-3D | 73.6 | 77.3 | **80.8** | 77.2 | 79.3 |
| Avg. | 70.1 | 74.1 | **75.1** | 74.7 | 75.7 |

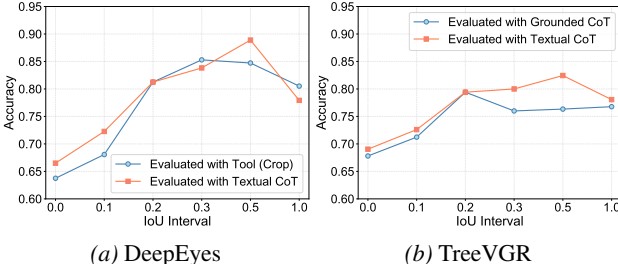

*(a) DeepEyes*      *(b) TreeVGR*

*Figure 2.* **Relationship between accuracy and localization quality (IoU).** Using HR8K (Wang et al., 2025e), questions are grouped based on the IoU of the generated grounded CoT, and accuracy is calculated for each IoU interval.

require the MLLM to generate bounding boxes when referring to objects in the CoT, thereby eliminating the need for external crop tools. However, the actual efficacy of such visually grounded CoT with crops or bounding boxes during inference lacks rigorous scrutiny.

To investigate the role of explicit crops or grounding in CoT, we conduct a comparative study using representative SOTA models, DeepEyes (Zheng et al., 2025b) and TreeVGR (Wang et al., 2025a). Specifically, we take off-the-shelf models originally trained with visually grounded CoT and evaluate them on multiple Visual Question Answering (VQA) benchmarks. Without any re-training, as reported in Table 1, we compare their default grounded CoT against a typical textual CoT, which is elicited simply by modifying the inference prompts (see Section A). For both models, inferring with visually grounded CoT outperforms the baseline Qwen2.5-VL-7B model (Bai et al., 2025). However, textual CoT achieves superior performance across most benchmarks, despite the models not being trained for text-only CoT. These results demonstrate that the capability of visually grounded reasoning can be implicitly transferred into typical textual CoT, obviating the need for explicit grounding in CoT.

To better understand the mechanism behind this observation, we examine the relationship between answer accuracy and localization quality (IoU, Intersection-over-Union) of the grounded CoT. Specifically, we manually label target bounding boxes for HR8K (Wang et al., 2025e) and group the questions based on the IoU of the generated grounded CoT. We then calculate the accuracy for each IoU interval, as illustrated in Figure 2. For DeepEyes (Zheng et al., 2025b) (see Figure 2a), visually grounded CoT with crops outperforms textual CoT when the crops are of high quality (i.e., IoU > 0.5). Conversely, when crop localization is poor, grounded CoT underperforms textual CoT. We hypothesize that inaccurate crops introduce visual noise,

which negatively impacts downstream answer prediction. For TreeVGR (Wang et al., 2025a) (see Figure 2b), however, visually grounded CoT consistently underperforms textual CoT across all IoU intervals. We hypothesize that the visual localization capability can be internalized into the textual CoT and that the mandatory task of explicit grounding imposes unnecessary interference, which detracts from the model's primary focus on answer prediction.

Building upon this insight, we propose **iVGR**, a reinforcement learning-based *dual-stream training strategy*. For each training query, the policy MLLM generates two distinct streams of rollouts: a *grounded stream* that incorporates explicit box predictions, and a *textual stream* that performs standard reasoning. To transfer the grounding capability across streams, we introduce a novel *consistency reward* that aligns the textual CoT with high-quality grounded reasoning trajectories selected from the grounded stream. In this way, the model internalizes localization into its textual reasoning without producing explicit coordinates at inference. Moreover, iVGR remains compatible with explicit crop tools at test time, and we further design a test-time workflow that exploits this synergy on fine-grained VQA. Extensive experiments on Qwen2.5-VL and Qwen3-VL demonstrate that iVGR yields significant improvements across multiple benchmarks.

Our contributions are summarized as follows:

- We analyze the efficacy of visually grounded reasoning, revealing that this capability can be implicitly transferred into typical textual CoT, thereby obviating the need for explicit grounding outputs during test time.

- Building on this insight, we propose **iVGR**, a reinforcement-learning-based dual-stream training strategy. By introducing a novel consistency reward, our method explicitly transfers the visually grounded reasoning capability into the textual reasoning process.

- We conduct extensive experiments on Qwen2.5-VL and Qwen3-VL models. The results demonstrate that

our models achieve significant improvements across multiple benchmarks and maintain compatibility with tool-assisted workflows during test time.

## 2. Related Work

**Reasoning MLLMs.** Chain-of-Thought (CoT) (Wei et al., 2022; Xu et al., 2025; Dong et al., 2025) has established itself as a cornerstone for enabling complex reasoning in both LLMs (Touvron et al., 2023; Yang et al., 2025; Liu et al., 2024) and MLLMs (Wang et al., 2025f; Huang et al., 2026; Lu et al., 2024). Notably, recent milestones such as DeepSeek-R1 (Guo et al., 2025) and OpenAI-o1 (Jaech et al., 2024) have demonstrated the effectiveness of RL post-training on scaling reasoning capabilities. Building on this momentum, advanced RL algorithms (Shao et al., 2024; Yu et al., 2025b; Zheng et al., 2025a) have proven effective in eliciting robust reasoning behaviors in language models. Subsequently, a wave of recent studies (Chen et al., 2025; Deng et al., 2025; Leng et al., 2025; Wang et al., 2025g; Huang et al., 2025; Yu et al., 2025a) has successfully adapted these RL-based paradigms to the multimodal domain.

**Visually Grounded Reasoning.** Although MLLMs have achieved remarkable success in general multimodal understanding, they continue to face significant challenges when processing high-resolution, fine-grained images (Wu & Xie, 2024; Wang et al., 2025e). To mitigate this, one line of research (Cao et al., 2025) focuses on enhancing local perception via cropping mechanisms. Early approaches like SEAL (Wu & Xie, 2024), Dyfo (Li et al., 2025), and ZoomEye (Shen et al., 2025) employ heuristic strategies to identify and crop regions of interest. TEVA (Jiang et al., 2025) incorporates off-the-shelf object detectors to locate informative details. More recently, inspired by OpenAI o3-style dynamic reasoning, RL-based methods such as Deep-Eyes (Zheng et al., 2025b), PixelReasoner (Wang et al., 2025c), and Mini-o3 (Lai et al., 2025) leverage GRPO training to enable models to dynamically invoke a crop tool during the reasoning process. Similarly, Thyme (Zhang et al., 2025b) extends this by generating code to execute diverse image-processing operations. In parallel, another line of research explores explicit visual grounding directly within the CoT, without relying on external tools. Methods such as GRIT (Fan et al., 2025) and TreeVGR (Wang et al., 2025a) require the model to predict bounding boxes for objects of interest, interleaved within the reasoning text. Specifically, GRIT designs reward signals for grounding in counting tasks, while TreeVGR generalizes this supervision to broader VQA scenarios. Distinct from these paradigms that rely on mandatory explicit tool calls or coordinate generation, our method aims to explicitly transfer and internalize this grounded reasoning capability into a pure textual

CoT. Furthermore, our method remains compatible with tool-assisted workflows to further push the envelope on fine-grained benchmarks.

## 3. Method

### 3.1. Preliminaries

**Textual *vs*. Visually Grounded CoT.** In our work, we distinguish between two kinds of CoT. The textual CoT, denoted as $\mathcal{Z}_{\text{text}}$, consists exclusively of natural language tokens describing the reasoning process. In contrast, the visually grounded CoT, denoted as $\mathcal{Z}_{\text{box}}$, explicitly incorporates visual localization. Formally, the visually grounded CoT contains bounding box predictions $\hat{b} = [x_1, y_1, x_2, y_2]$, allowing the model to explicitly anchor its reasoning on specific image regions. Although these bounding boxes are technically represented as natural language coordinates, we classify this format distinctively to highlight the explicit grounding mechanism. Additionally, the grounded CoT can optionally leverage crop tools with the predicted bounding boxes in the reasoning process.

### 3.2. Dual-Stream Training Strategy

As discussed in Section 1, empirical studies suggest that the off-the-shelf grounded reasoning models (e.g., Deep-Eyes (Zheng et al., 2025b), TreeVGR (Wang et al., 2025a)) often underperform with explicit grounded CoT compared to typical textual CoT during inference. Our further analysis reveals that textual CoT consistently achieves superior accuracy across various localization quality intervals when no external tools are invoked. We hypothesize that visual localization capabilities can be effectively internalized into the textual CoT; conversely, the mandatory generation of explicit coordinates introduces task interference, detracting from the model's primary objective of answer prediction.

Building on this insight, we propose **iVGR**, a reinforcement-learning-based dual-stream training strategy designed to explicitly transfer visual localization capabilities into the textual reasoning process. As illustrated in Figure 3, for each training query $q$, the policy MLLM $\pi_\theta$ generates two distinct streams of rollouts conditioned on stream-specific prompts: a grounded stream and a textual stream. The grounded stream requires the model to explicitly predict bounding boxes when referring to objects, whereas the textual stream elicits standard natural language reasoning without explicit grounding constraints.

During training, we sample a group of $N$ rollouts for each stream: $\mathcal{O}^b = \{o_1^b, \ldots, o_N^b\}$ for the grounded stream, and $\mathcal{O}^t = \{o_1^t, \ldots, o_N^t\}$ for the textual stream. For the grounded stream, we adopt the reward formulation from TreeVGR (Wang et al., 2025a), incorporating format, answer accuracy, and localization quality. For the textual

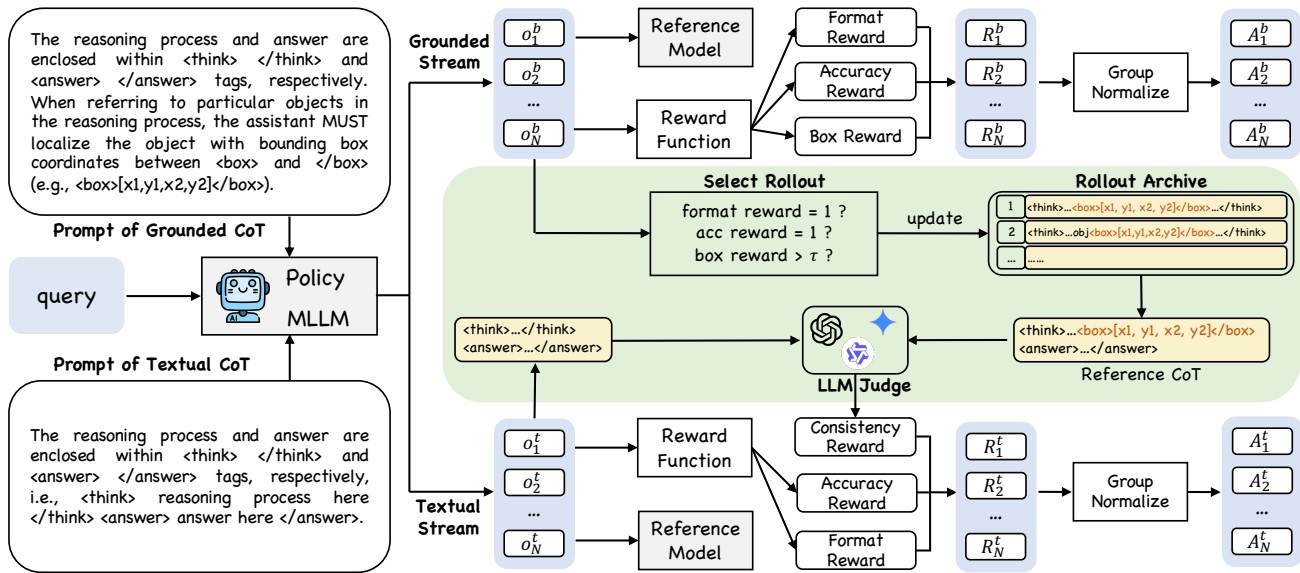

*Figure 3.* **Illustration of the proposed method.** We employ a dual-stream training strategy consisting of a grounded stream and a textual stream. The grounded stream mandates the prediction of bounding box coordinates within the CoT when referring to objects. In parallel, the textual stream performs standard natural language reasoning, where a consistency reward is introduced to align its semantic logic with high-quality grounded references selected from the grounded stream.

stream, alongside standard format and accuracy rewards, we introduce a novel consistency reward to enforce the internalization of visually grounded reasoning. Finally, we compute advantages for each stream, $\mathcal{A}^b = \{A_1^b, \ldots, A_N^b\}$ and $\mathcal{A}^t = \{A_1^t, \ldots, A_N^t\}$, using group-wise normalization following the GRPO framework.

**Grounded Stream.** We first sample $N$ rollouts $\mathcal{O}^b$ from the policy $\pi_\theta$ using a prompt that enforces explicit bounding box prediction within the CoT. The total reward for the $i$-th rollout $o_i^b$ is defined as:

$$R_i^b = R_{\text{format}} + R_{\text{acc}} + R_{\text{box}}. \quad (1)$$

The components are defined as follows:

- **Format Reward** ($R_{\text{format}}$): We require the output to strictly follow the `<think>...</think>` `<answer>...</answer>` structure. We assign $R_{\text{format}} = 1$ for valid rollouts, and 0 otherwise.

- **Accuracy Reward** ($R_{\text{acc}}$): This binary reward indicates correctness. $R_{\text{acc}} = 1$ if the predicted answer matches the ground truth, else 0.

- **Box Reward** ($R_{\text{box}}$): To evaluate localization quality between the set of predicted boxes $\mathcal{B}_{\text{pred}}$ and ground-truth boxes $\mathcal{B}_{\text{gt}}$, we employ a bidirectional IoU matching metric. Formally, let $\text{MaxIoU}(b, \mathcal{B}) = \max_{b' \in \mathcal{B}} \text{IoU}(b, b')$. The box reward is defined as the average of the recall-oriented and precision-oriented

matching scores:

$$R_{\text{box}} = \frac{1}{2} \left( \frac{1}{|\mathcal{B}_{\text{gt}}|} \sum_{b \in \mathcal{B}_{\text{gt}}} \text{MaxIoU}(b, \mathcal{B}_{\text{pred}}) \right.$$
$$\left. + \frac{1}{|\mathcal{B}_{\text{pred}}|} \sum_{\hat{b} \in \mathcal{B}_{\text{pred}}} \text{MaxIoU}(\hat{b}, \mathcal{B}_{\text{gt}}) \right). \quad (2)$$

Advantages $A^b$ are then computed by normalizing $R^b$ using the group's mean and standard deviation.

**Textual Stream.** Simultaneously, we sample $N$ rollouts $\mathcal{O}^t$ using a standard prompt that requests only reasoning and the answer. The total reward for the $i$-th rollout $o_i^t$ is formulated as:

$$R_i^t = R_{\text{format}} + R_{\text{acc}} + R_{\text{consistency}}, \quad (3)$$

where $R_{\text{consistency}}$ measures the semantic alignment between the textual CoT and a high-quality grounded reference (detailed in Section 3.3). Note that $R_{\text{format}}$ and $R_{\text{acc}}$ are calculated identically to those in the grounded stream. Advantages $A^t$ are similarly derived via group normalization.

### 3.3. Consistency Reward

We propose the consistency reward to transfer visual localization capabilities from the grounded stream to the textual stream. This is achieved by treating high-quality grounded CoTs as visual guides and encouraging the textual CoT to

maintain semantic consistency with them. The process involves three key components: Reference Selection, Rollout Archive Maintenance, and LLM-based Consistency Scoring.

**Reference Selection and Rollout Archive.** To ensure the textual stream learns from correct visual perceptions, we must filter out low-quality grounded rollouts. For a grounded rollout $o^b$ to be considered a valid reference, it must satisfy three criteria: (1) valid format ($R_{\text{format}} = 1$), (2) correct answer ($R_{\text{acc}} = 1$), and (3) high-quality localization ($R_{\text{box}} > \tau$, where $\tau$ is a predefined IoU threshold).

Since the policy is updated dynamically, the quality of grounded rollouts may fluctuate within a single batch. To stabilize training, we maintain a Rollout Archive that persists the optimal grounded CoT found so far for each training sample. Specifically, let $\mathcal{Z}_{\text{archive}}^{(q)}$ denote the archived grounded CoT for query $q$. At each training step, we identify the best valid rollout $o_{\text{best}}^b$ from the current group $\mathcal{O}^b$ (i.e., the one with the highest $R_{\text{box}}$ among valid rollouts). We then update the archive if the new rollout is superior:

$$\mathcal{Z}_{\text{archive}}^{(q)} \leftarrow \arg \max_{z \in \{\mathcal{Z}_{\text{archive}}^{(q)}, o_{\text{best}}^b\}} R_{\text{box}}(z). \quad (4)$$

This mechanism ensures that the consistency reward is always computed against the highest-quality visual guides available.

**LLM-based Consistency Scoring.** Given a textual rollout $o^t$ and the retrieved reference $\mathcal{Z}_{\text{archive}}^{(q)}$, we employ an external LLM (e.g., Qwen2.5-72B (Yang et al., 2024)) as a judge to quantify their semantic consistency. The judge evaluates whether the textual CoT describes the same visual content as the grounded reference. We define a granular scoring rubric $\alpha = \mathcal{S}(o^t, \mathcal{Z}_{\text{archive}}^{(q)})$ as follows:

- $\alpha = 1.0$: The textual CoT describes image content identical to the reference.

- $\alpha = 0.7$: The textual CoT exhibits a single type of deviation: it either omits details present in the reference or introduces new details not found in the reference, but not both.

- $\alpha = 0.3$: The textual CoT exhibits compound deviations: it simultaneously omits information present in the reference and introduces extraneous details, indicating a significant divergence in focus.

- $\alpha = 0.0$: The textual CoT contains descriptions that explicitly contradict the visual facts established in the reference.

The final consistency reward is then $R_{\text{consistency}} = \alpha$. If the archive is empty (i.e., no valid grounded rollout has been found yet), we set $R_{\text{consistency}} = 0$.

### 3.4. Tool-Assisted Test-Time Scaling

Although our model is optimized to internalize visually grounded reasoning, the dual-stream training preserves its explicit capability to generate bounding boxes when prompted. To leverage this flexibility for handling fine-grained details, we devise a tool-assisted test-time scaling workflow that synergizes the model's grounding capability with external cropping tools. The workflow proceeds in three steps: First, we prompt the model to generate a grounded CoT and extract all predicted bounding boxes. Second, we utilize a crop tool to generate a set of local views based on these coordinates. Crucially, to preserve the spatial context and relative relationships among objects, we additionally construct a union crop, which is defined as the minimum enclosing rectangle covering all predicted boxes. Finally, we aggregate the original global image, the individual object crops, and the union crop as multi-view visual inputs, and prompt the model to perform reasoning using standard textual CoT.

## 4. Experiments

### 4.1. Setup

**Datasets.** Our training pipeline consists of two distinct stages: a cold-start supervised fine-tuning (SFT) phase and a reinforcement learning (RL) phase. **Cold-Start Stage:** To initialize the ability to generate both valid grounded formats and high-quality textual reasoning, we utilize the TreeVGR-SFT-35K dataset (Wang et al., 2025a). Specifically, we partition this dataset into two subsets: we retain 32,000 samples with their original grounded CoT annotations, and for the remaining 3,000 samples, we employ Qwen2.5-VL-72B-Instruct to synthesize pure textual CoT responses. This results in a mixed cold-start dataset comprising 32K grounded samples and 3K textual samples. **RL Stage:** We utilize the TreeVGR-RL-37K dataset, which aggregates 32,000 samples from V* (Wu & Xie, 2024) and 5,000 samples from VisDrone (Zhu et al., 2021). These instances provide ground-truth bounding-box annotations, facilitating the computation of box rewards ($R_{\text{box}}$) and consistency rewards. Furthermore, to enhance generalization across chart understanding and multidisciplinary reasoning, we incorporate an additional 14,000 samples from OpenMMReasoner (Zhang et al., 2025a) and ArxivQA (Li et al., 2024b). Since these supplementary samples lack bounding-box annotations, we exclusively utilize them to train the textual stream, omitting the computation of the consistency reward for this subset. More details about training data and implementation are shown in Section B.

**Evaluation Benchmarks.** We conduct evaluations across a diverse set of datasets. First, consistent with existing visually grounded reasoning studies, we evalu-

*Table 2.* **Performance on fine-grained and general VQA benchmarks.** All results are obtained using VLMEvalKit (Duan et al., 2024).

| Models | Tools | Fine-grained VQA | | | General VQA | | | | | Avg. |
|---|---|---|---|---|---|---|---|---|---|---|
| | | V* | HR4K | HR8K | MME RW-Lite | POPE | RealWorld QA | CV-2D | CV-3D | |
| **Proprietary Models** | | | | | | | | | | |
| Gemini-3.1-Pro-Preview | - | 87.4 | 88.9 | 88.1 | 55.8 | 88.0 | 83.5 | 85.0 | 94.6 | 83.9 |
| GPT-5.4 | - | 88.0 | 87.4 | 80.6 | 63.4 | 87.9 | 83.0 | 82.4 | 91.9 | 83.1 |
| **Open-source General Models** | | | | | | | | | | |
| LLaVA-OneVision-7B (Li et al., 2024a) | ✗ | 72.8 | 64.6 | 57.9 | 48.2 | 88.3 | 69.5 | 72.9 | 76.9 | 68.9 |
| InternVL3-8B (Zhu et al., 2025) | ✗ | 70.2 | 70.0 | 69.3 | 48.6 | 90.3 | 71.0 | 80.6 | 86.1 | 73.3 |
| Qwen2.5-VL-7B (Bai et al., 2025) | ✗ | 78.5 | 69.0 | 65.1 | 44.5 | 86.3 | 68.1 | 75.7 | 73.6 | 70.1 |
| Qwen2.5-VL-32B (Bai et al., 2025) | ✗ | 80.1 | 73.0 | 69.5 | 46.3 | 86.5 | 70.1 | 76.7 | 84.5 | 73.3 |
| Qwen2.5-VL-72B (Bai et al., 2025) | ✗ | 85.9 | 79.9 | 76.8 | 45.2 | 86.3 | 76.1 | 78.4 | 87.2 | 77.0 |
| Qwen3-VL-4B (Yang et al., 2025) | ✗ | 78.5 | 77.8 | 71.1 | 48.3 | 89.3 | 71.2 | 78.7 | 91.7 | 75.8 |
| Qwen3-VL-8B (Yang et al., 2025) | ✗ | 82.7 | 76.5 | 70.4 | 49.0 | 88.1 | 70.5 | 78.6 | 93.5 | 76.2 |
| Qwen3-VL-32B (Yang et al., 2025) | ✗ | 83.8 | 80.0 | 78.1 | 52.1 | 89.4 | 79.3 | 81.2 | 92.8 | 79.6 |
| **Visually Grounded Reasoning Models** | | | | | | | | | | |
| GRIT-3B (Fan et al., 2025) | ✗ | 54.5 | 48.4 | 43.5 | 33.8 | 80.8 | 58.0 | 72.5 | 68.2 | 57.5 |
| PixelReasoner-7B (Wang et al., 2025c) | ✓ | - | 72.9 | 66.9 | 49.7 | - | - | - | - | - |
| DeepEyes-7B (Zheng et al., 2025b) | ✓ | 82.7 | 75.1 | 72.6 | 53.2 | 87.7 | 69.4 | 75.0 | 77.3 | 74.1 |
| DeepEyesV2-7B (Hong et al., 2025) | ✓ | 81.8 | 77.9 | 73.8 | - | - | - | - | - | - |
| Mini-o3-7B (Lai et al., 2025) | ✓ | - | 77.5 | 73.3 | - | - | - | - | - | - |
| Thyme-7B (Zhang et al., 2025b) | ✓ | 82.2 | 77.0 | 72.0 | 55.2 | 86.8 | **70.2** | 78.0 | 75.1 | 74.6 |
| TreeVGR-7B (Wang et al., 2025a) | ✗ | 83.8 | 77.1 | 73.1 | 54.9 | 87.3 | 67.3 | 76.6 | 77.2 | 74.7 |
| iVGR-Qwen2.5-VL-7B (**ours**) | ✗ | **86.4** | **78.3** | **75.5** | **55.6** | **88.9** | 68.6 | **78.4** | **81.1** | **76.6** |
| Δ *v.s.* Qwen2.5-VL-7B | | +7.9 | +9.3 | +10.4 | +11.1 | +2.6 | +0.5 | +2.7 | +7.5 | +6.5 |
| iVGR-Qwen3-VL-8B (**ours**) | ✗ | 90.1 | 82.0 | 80.1 | 60.7 | 89.4 | 71.0 | 80.8 | 91.0 | 80.6 |
| Δ *v.s.* Qwen3-VL-8B | | +7.4 | +5.5 | +9.7 | +11.7 | +1.3 | +0.5 | +2.2 | -2.5 | +4.4 |
| iVGR-Qwen3-VL-32B (**ours**) | ✗ | 93.2 | 82.9 | 82.9 | 61.2 | 88.8 | 76.3 | 83.9 | 93.8 | 82.9 |
| Δ *v.s.* Qwen3-VL-32B | | +9.4 | +2.9 | +4.8 | +9.1 | -0.6 | -3.0 | +2.7 | +1.0 | +3.3 |

ate fine-grained perception on V* (Wu & Xie, 2024), HR4K, and HR8K (Wang et al., 2025e). Furthermore, we extend our evaluation to general VQA (MME-RW-Lite (Zhang et al., 2024), POPE (Li et al., 2023), RealWorldQA, CV-2D/3D (Tong et al., 2024)), chart understanding (ChartQA (Masry et al., 2022), AI2D (Kembhavi et al., 2016)), and multidisciplinary reasoning (WeMath (Qiao et al., 2025), MMStar (Chen et al., 2024), MMMU (Yue et al., 2024), MMK12 (Meng et al., 2025)).

**Implementation Details.** We use Llama-Factory (Zheng et al., 2024) and VeRL (Sheng et al., 2024) frameworks to conduct SFT and RL training, respectively. More implementation details are shown in Section C.

### 4.2. Main Results

As summarized in Table 2, we evaluate our method against leading baselines on both fine-grained and general VQA benchmarks. Building on the Qwen2.5-VL-7B, our method consistently outperforms existing visually grounded approaches, surpassing both tool-based and explicit grounding-based methods. Notably, our method demonstrates substantial advantages on fine-grained tasks: it outperforms pre-

vious state-of-the-art models by 2.6% on V* and 1.7% on HR8K, validating the efficacy of our internalized grounding strategy. On general VQA benchmarks, our method maintains this superiority. To verify scalability, we extend our experiments to the larger Qwen3-VL-8B and Qwen3-VL-32B models, achieving average improvements of 4.4% and 3.3%, respectively. These experimental results confirm that our method scales effectively across different model sizes. We further assess the generalization capability of our method on diverse visual domains, including chart understanding and multidisciplinary reasoning tasks. As reported in Table 3, iVGR yields significant and consistent gains over baseline models across multiple benchmarks.

### 4.3. Results of Tool-Assisted Test-Time Scaling

We further evaluate our models' ability to scale at test time with the crop tool on fine-grained benchmarks. As reported in Table 4, when equipped with crops from the predicted bounding boxes, our models can be enhanced by 1.1% and 1.5% on HR4K based on Qwen2.5-VL-7B and Qwen3-VL-8B, respectively. To further enhance spatial context and relationships between predicted objects, we supplement the

*Table 3.* **Evaluation results on chart understanding and multidisciplinary reasoning benchmarks.**

| Models | Chart Understanding | | Multidisciplinary Reasoning | | | | Avg. |
|---|---|---|---|---|---|---|---|
| | $\text{ChartQA}_{test}$ | $\text{AI2D}_{test}$ | WeMath | MMStar | $\text{MMMU}_{val}$ | MMK12 | |
| Qwen2.5-VL-7B | 86.4 | 83.6 | 35.3 | 63.9 | 54.4 | 53.6 | 62.9 |
| iVGR-Qwen2.5-VL-7B (**ours**) | **88.5** | **85.0** | **41.1** | **66.3** | **55.2** | **56.3** | **65.4** (+2.5) |
| Qwen3-VL-8B | 83.2 | 80.4 | 49.7 | 67.9 | 58.0 | 60.4 | 66.6 |
| iVGR-Qwen3-VL-8B (**ours**) | **87.6** | **85.5** | **55.1** | **69.7** | **59.8** | **61.6** | **69.9** (+3.3) |
| Qwen3-VL-32B | 85.0 | 84.5 | 60.0 | 72.3 | **67.7** | 73.9 | 73.9 |
| iVGR-Qwen3-VL-32B (**ours**) | **90.4** | **88.7** | **61.6** | **75.1** | **67.7** | **75.2** | **76.5** (+2.6) |

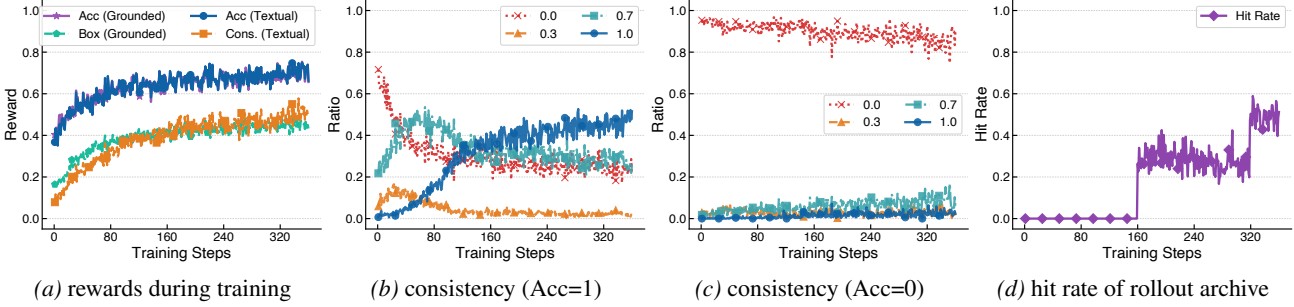

*(a) rewards during training*    *(b) consistency (Acc=1)*    *(c) consistency (Acc=0)*    *(d) hit rate of rollout archive*

*Figure 4.* **Analysis of training dynamics.** (a) Evolution of accuracy and specific rewards (box reward for the grounded stream; consistency reward for the textual stream) throughout the training process. (b-c) Distribution of consistency scores among (b) correctly predicted samples and (c) incorrectly predicted samples. (d) The hit rate of the rollout archive, representing the frequency with which historical best rollouts are utilized as references.

*Table 4.* **Evaluation results on high-resolution fine-grained VQA benchmarks.** The suffix "+ crops" indicates the integration of local views extracted from predicted bounding boxes, while "+ union crop" denotes the inclusion of a context view defined by the minimum enclosing rectangle covering all predicted regions.

| Models | V* | HR4K | HR8K | Avg. |
|---|---|---|---|---|
| Qwen2.5-VL-7B | 78.5 | 69.0 | 65.1 | 70.9 |
| iVGR-7B | 86.4 | 78.3 | 75.5 | 80.1 |
| iVGR-7B + crops | 89.0 | 79.4 | **76.3** | 81.6 |
| iVGR-7B + union crop | 89.0 | 79.9 | 75.8 | 81.6 |
| iVGR-7B + crops + union crop | **90.1** | **81.8** | **76.3** | **82.7** |
| Qwen3-VL-8B | 82.7 | 76.5 | 70.4 | 76.5 |
| Qwen3-VL-8B + tool | 90.1 | 82.3 | 78.0 | 83.5 |
| iVGR-8B | 90.1 | 82.0 | **80.1** | 84.1 |
| iVGR-8B + crops | 89.5 | 83.5 | 78.0 | 83.7 |
| iVGR-8B + union crop | 92.7 | **84.5** | 78.8 | 85.3 |
| iVGR-8B + crops + union crop | **93.2** | 84.3 | 79.3 | **85.6** |

union crop of all predicted boxes as an additional local view, which enhances our models by 3.5% and 2.3% on HR4K based on Qwen2.5-VL-7B and Qwen3-VL-8B, respectively. Experimental results demonstrate the effectiveness of the proposed tool-assisted test-time scaling workflow.

### 4.4. Ablation Study

**Analysis of Training Dynamics.** To investigate the optimization process of iVGR, we visualize the evolution of key training metrics in Figure 4. Focusing on the consistency re-

ward distribution for correctly answered queries (Figure 4b), we observe a clear convergence trend: the proportion of fully consistent rollouts (Score 1.0) steadily increases to dominance (~50%) by the end of training, while contradictory rollouts (Score 0.0) significantly decline. Interestingly, the intermediate scores (0.3 and 0.7) exhibit a distinct "rise-and-fall" pattern. This trend suggests a progressive learning curriculum, where the model transitions from complete hallucination (0.0) to partial alignment (0.3/0.7) before ultimately converging to high-fidelity alignment (1.0). In parallel, we examine the utilization of the rollout archive in Figure 4d. As expected, the hit rate remains zero during the initial epoch (steps 0-160) while the archive is being populated. Subsequently, it jumps to ~30% in the second epoch and climbs to ~50% in the third epoch. This growing utilization confirms that the archive effectively stabilizes training by providing an expanding pool of high-quality historical grounded CoT for consistency evaluation.

**Ablation of Components.** Table 5 presents a component-wise ablation study of our method. To ensure a fair comparison, all RL variants are based on the same cold-start model (variant 2), which is fine-tuned on a mixture of grounded and textual CoT data. We first examine the impact of single-stream RL training. Training exclusively with the grounded stream but evaluating with textual CoT (variant 4) yields superior performance compared to training purely on the textual stream (variant 3). Next, we observe that simply combining both streams without the consistency reward (variant

*Table 5.* **Ablation study of components in our method.** 'T' and 'G' denote the textual CoT and grounded CoT during test time, respectively. All variants (3)-(7) are based on the same cold-start model, which is trained on the mixture of grounded CoT and textual CoT.

| No. | Variants | CoT | V* | HR4K | HR8K | MME-RW-Lite | MMStar | Avg. |
|---|---|---|---|---|---|---|---|---|
| (1) | Qwen2.5-VL-7B | T | 78.5 | 69.0 | 65.1 | 44.5 | 63.9 | 64.2 |
| (2) | + Cold-Start | T | 80.1 | 73.2 | 67.5 | 49.5 | 65.0 | 67.1 |
| | | G | 80.1 | 72.5 | 66.8 | 49.5 | 63.8 | 66.5 |
| (3) | + GRPO w/ textual CoT (textual stream only) | T | 85.9 | 75.5 | 68.6 | 50.7 | 64.9 | 69.1 |
| (4) | + GRPO w/ grounded CoT (grounded stream only) | T | **87.4** | 76.0 | 71.8 | 54.3 | 66.2 | 71.1 |
| | | G | 86.4 | 76.1 | 72.1 | 54.1 | 65.1 | 70.8 |
| (5) | + Dual-Stream Training w/o $R_{\text{consistency}}$ | T | **87.4** | 75.3 | 72.4 | 55.0 | 64.7 | 71.0 |
| (6) | + Dual-Stream Training w/ $R_{\text{consistency}}$ | T | 86.4 | 77.3 | 73.2 | 55.1 | 65.5 | 71.5 |
| (7) | + Dual-Stream Training w/ $R_{\text{consistency}}$ + Rollout Archive (**ours**) | T | 86.4 | **78.3** | **75.5** | **55.6** | **66.3** | **72.4** |
| | | G | 85.9 | 77.1 | 74.3 | 55.3 | 65.1 | 71.5 |

*Table 6.* **Influence of threshold $\tau$ for grounded CoT selection.**

| $\tau$ | 0.0 | 0.1 | 0.2 | 0.3 | 0.4 |
|---|---|---|---|---|---|
| V* | 86.4 | 86.9 | **87.4** | 86.4 | **87.4** |
| HR4K | 76.1 | 78.0 | 78.0 | **78.3** | 75.4 |
| HR8K | 73.5 | 73.5 | 74.1 | **75.5** | 72.2 |
| MME-RW-Lite | **55.8** | 54.6 | 55.4 | 55.6 | 53.6 |
| MMStar | 66.1 | 64.5 | **66.7** | 66.3 | 64.8 |
| **Avg.** | 71.6 | 71.5 | 72.3 | **72.4** | 70.7 |

*Table 7.* **Impact of different LLM judge models for consistency reward.** We compare the performance when using different Qwen2.5 models (14B, 32B, 72B) to judge the consistency reward.

| Model Sizes | 14B | 32B | 72B |
|---|---|---|---|
| V* | 86.4 | 86.4 | 86.4 |
| HR4K | 76.6 | **79.0** | 78.3 |
| HR8K | 73.9 | 73.0 | **75.5** |
| MME-RW-Lite | 54.4 | 54.8 | **55.6** |
| MMStar | **66.3** | 65.1 | **66.3** |
| **Avg.** | 71.5 | 71.7 | **72.4** |

*Table 8.* **Influence of reference CoT localization quality on the consistency reward.** We use grounded CoTs from a specified box IoU interval as the reference CoT for the consistency reward.

| IoU Interval | V* | HR4K | HR8K | MME-RW-Lite | MMStar | Avg. |
|---|---|---|---|---|---|---|
| w/o $R_{\text{consistency}}$ | **87.4** | 75.3 | 72.4 | 55.0 | 64.7 | 71.0 |
| $[0, 0.1)$ | 86.9 | 76.5 | 74.0 | 55.0 | 65.3 | 71.5 |
| $[0.1, 0.3)$ | 86.9 | 77.8 | 74.1 | **55.7** | 64.9 | 71.9 |
| $[0.3, 1]$ (**ours**) | 86.4 | **78.3** | **75.5** | 55.6 | **66.3** | **72.4** |

however, this strict filtering results in sparse supervision, as fewer training samples can retrieve a valid reference to compute the consistency reward. Conversely, employing a lower $\tau$ increases the coverage of usable references but inevitably introduces noisy supervision from low-quality grounding, which may mislead the textual stream. We set $\tau$ to 0.3 in our experiments.

**Influence of Different LLM Judge Models.** As reported in Table 7, we compare the performance of our method when employing judge models of varying sizes. Experimental results indicate that scaling up the judge model yields superior performance. This positive correlation demonstrates that the accuracy of the consistency reward is critical for the effectiveness of our method.

**Influence of the Quality of Reference CoTs.** We analyze how the localization quality of reference CoTs affects the consistency reward. First, qualified reference CoTs are abundant during training. We find that 77.3% of queries eventually obtain a reference CoT with IoU $\geq 0.3$, while only 11.4% and 11.3% remain in the $[0, 0.1)$ and $[0.1, 0.3)$ intervals, respectively. For queries that lack any qualified reference, the consistency reward is strictly set to 0 to prevent misleading supervision of the textual stream. Second, the performance of our method is positively correlated with the localization quality of reference CoTs. As shown in Table 8, we use grounded CoTs from different box IoU inter-

5) performs comparably to the grounded-only baseline, suggesting that the dual-stream architecture alone offers limited gains. Crucially, introducing the proposed consistency reward (variant 6) significantly boosts performance, achieving a 7.3% improvement over the Qwen2.5-VL-7B baseline. Finally, incorporating the rollout archive (variant 7) further elevates the average performance to 72.4, resulting in a total gain of 8.2%. These results validate that the rollout archive effectively stabilizes training by maintaining high-quality references, thereby maximizing the efficacy of the consistency alignment.

**Influence of Threshold $\tau$.** We ablate the influence of the hyper-parameter $\tau$, which governs the selection of qualified reference CoTs, in Table 6. Using a higher $\tau$ ensures that only high-quality grounded CoTs serve as references;

*Table 9.* **Comparison between a single judge score and the average of four sampled judge scores.** 'Single score' uses Qwen2.5-72B to produce a single judge score with the sampling temperature set to 0.01, whereas 'avg. of four scores' uses Qwen2.5-72B to judge four times (sampling temperature set to 0.5) and takes the average of the four scores as the final result.

| Strategy | V* | HR4K | HR8K | MME-RW-Lite | POPE | RealWorldQA | CV-2D | CV-3D | **Avg.** |
|---|---|---|---|---|---|---|---|---|---|
| Single score | 86.4 | **78.3** | **75.5** | **55.6** | 88.9 | 68.6 | **78.4** | 81.1 | 76.6 |
| Avg. of four scores | **88.5** | 77.4 | 74.4 | 54.7 | **89.1** | **70.5** | 78.1 | **82.2** | **76.9** |

*Table 10.* **Attention scores on image tokens at the final transformer layer.**

| Variant | HR4K | HR8K | MME-RW-Lite |
|---|---|---|---|
| Textual stream only | 0.0959 | 0.0960 | 0.1107 |
| Grounded stream only | 0.1113 | 0.1115 | 0.1258 |
| iVGR (textual CoT, **ours**) | **0.1239** | **0.1235** | **0.1320** |

vals as the reference CoT, and our default setting achieves the best results. Importantly, even with the lowest interval $[0, 0.1)$, our method still outperforms the variant without a consistency reward, demonstrating the effectiveness of our design.

**Internalization of Visual Grounding.** To examine whether iVGR truly internalizes the visual localization capability into the textual reasoning process, we measure how strongly the model attends to the image during inference. Specifically, for each generated CoT, we compute the attention score assigned to image tokens at the final transformer layer, and report the results in Table 10. iVGR with textual CoT yields the highest attention on image tokens across all three benchmarks, even surpassing the grounded-stream-only variant that produces explicit coordinates. This indicates that the consistency reward effectively transfers the visual grounding capability from the grounded stream into the textual stream, allowing the model to maintain a strong awareness of visual evidence without emitting explicit coordinates at inference.

## 5. Conclusion and Limitations

In this work, we introduce **iVGR**, a reinforcement learning framework designed to internalize visual localization capabilities into the textual reasoning process. Central to our approach is a dual-stream training strategy, which utilizes a consistency reward to enforce semantic alignment between the textual CoT and a high-quality grounded reference rollout. Extensive experimental results demonstrate the effectiveness and scalability of our method, showing consistent improvements across different model sizes. Furthermore, we devise a tool-integrated test-time scaling workflow, demonstrating iVGR's flexibility in synergizing its intrinsic grounding capability with external tools to further boost fine-grained perception performance.

**Limitations and Future Work.** We discuss limitations and future work in this section. First, our method requires bounding-box annotations, which imposes additional demands on data labeling. Nevertheless, we believe this burden can be substantially alleviated through two complementary strategies: (1) repurposing existing detection datasets (e.g., for the counting task), and (2) leveraging off-the-shelf detectors for pseudo-labeling. These strategies significantly reduce human effort. Furthermore, our method is compatible with general textual reasoning tasks. As shown in Table 12, the absence of bounding boxes in certain tasks does not negatively impact our model's performance, demonstrating its broad applicability.

Second, our method relies on an external LLM to score the consistency between rollouts and reference CoTs. Since LLMs may produce hallucinations, the accuracy of the consistency reward can be affected, which remains a challenging problem. We can improve the quality of LLM judge scores by scaling the judge model and ensembling multiple judge scores. (1) *Scaling the judge model*: In Table 7, we ablate the influence of different judge models across various model sizes. Experimental results demonstrate that a larger judge model brings greater benefits to our method. We believe that a stronger judge model can produce higher-quality scores and further improve the performance of our method. (2) *Ensembling multiple judge scores*: We can improve the quality of judge scores by ensembling multiple sampling results. Specifically, we adopt Qwen2.5-72B as the judge model and set the sampling temperature to 0.5. We then sample four judge scores for each textual CoT and average them as the consistency reward. As shown in Table 9, experimental results demonstrate that this approach further improves the overall performance of our method.

Third, since our method does not employ a crop tool during training, the grounded stream may hallucinate or miss details when handling very high-resolution images or extremely fine-grained content, which in turn limits the quality of the reference CoTs available for the consistency reward. As shown in Table 4, our method can be further enhanced when equipped with a crop tool at test time. Building on this observation, we believe the grounded stream in our framework can be replaced by a crop-tool-augmented reasoning mechanism during training, which would yield more accurate and detail-rich reference CoTs and thus stronger supervision for the textual stream.

## Impact Statement

This paper presents work whose goal is to advance the field of Machine Learning. There are many potential societal consequences of our work, none of which we feel must be specifically highlighted here.

## Acknowledgements

This work is supported by Hong Kong Research Grants Council – General Research Fund (Grant No. 17211024), Hong Kong Innovation and Technology Commission – Innovation and Technology Fund (Project No. ITS/488/24FP), and HKU Seed Fund for PI Research.

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

# A. Prompts

In this section, we introduce the detailed prompts used in our work. We begin with the evaluation prompt used for the off-the-shelf DeepEyes (Zheng et al., 2025b) and TreeVGR (Wang et al., 2025a) models, utilizing textual CoT.

---

**System Prompt for Evaluation**

To answer the question, you **MUST** think first in your mind carefully between `<think>`...`</think>`, then carefully give the correct answer between `<answer>`...`</answer>`.
The output is formatted strictly as: `<think>`...`</think>` `<answer>`...`</answer>`.
Please make sure the answer is correct and concise.

---

Next, we present the training prompts employed for the grounded stream and the textual stream, respectively.

---

**Prompt for the Grounded Stream Training**

**System prompt:**
You are a helpful assistant.

**User prompt:**
**{query}**
A conversation between user and assistant. The user asks a question, and the Assistant solves it. The assistant **MUST** first think about the reasoning process in the mind and then provide the user with the answer.
The reasoning process and answer are enclosed within `<think>` `</think>` and `<answer>` `</answer>` tags, respectively.
When referring to particular objects in the reasoning process, the assistant **MUST** localize the object with bounding box coordinates between `<box>` and `</box>` (e.g., `<box>`[x1,y1,x2,y2]`</box>`). You **MUST** strictly follow the format.

**Example:**
**User:** What color is the ball on the left?
**Assistant:** `<think>`There are multiple balls in the image. To accurately refer to the one on the left, I localize it with a bounding box. The ball `<box>`[50,80,120,150]`</box>` within this region is blue.`</think>` `<answer>`Blue`</answer>`

---

**Prompt for the Textual Stream Training**

**System prompt:**
You are a helpful assistant.

**User prompt:**
**{query}**
A conversation between User and Assistant. The user asks a question, and the Assistant solves it. The assistant first thinks about the reasoning process in the mind and then provides the user with the answer.
The reasoning process and answer are enclosed within `<think>` `</think>` and `<answer>` `</answer>` tags, respectively, i.e., `<think>` reasoning process here `</think>` `<answer>` answer here `</answer>`.

---

The prompt employed for the consistency reward judge model is presented below. It takes the original question, the reference grounded CoT, and the target textual CoT as inputs. The judge model is tasked with directly evaluating the semantic consistency score between the two CoTs, thereby minimizing inference overhead.

---

**Prompt for Consistency Reward Judge Model**

**System prompt:**
You are a helpful assistant.

**User prompt:**
You are an expert at evaluating visual descriptions in reasoning chains.
I will provide two reasoning processes for the same visual question:

- **Reference CoT**: A reasoning chain with visual grounding (bounding boxes). Assume its image descriptions are CORRECT.

- **Target CoT**: A reasoning chain without bounding boxes.

Your task is to compare the **image content descriptions** in Target CoT against Reference CoT.

**Scoring Rules (mutually exclusive, check in order):**

**1. Score 0.0:** Target CoT contains **ANY** contradiction or conflict with Reference CoT's image descriptions.

**2. Score 0.3:** Target CoT has **NO** contradiction, BUT it **BOTH**:
- Contains descriptions/details **NOT** present in Reference CoT, **AND**
- Misses some descriptions/details that **ARE** present in Reference CoT.

**3. Score 0.7:** Target CoT has **NO** contradiction, BUT **ONE** of the following:
- Contains descriptions/details **NOT** present in Reference CoT, **OR**
- Misses some descriptions/details that **ARE** present in Reference CoT.

**4. Score 1.0:** Target CoT's image descriptions are **FULLY CONSISTENT** with Reference CoT – no contradiction, no extra details, no missing details.

**Input:**
**Question**: `{question}`
**Reference CoT (with boxes, assume correct):** `{reference_think}`
**Target CoT (without boxes):** `{target_think}`
**Output:**
Output ONLY the score (0.0, 0.3, 0.7, or 1.0), nothing else.

Score:

---

## B. Training Data

Our training data comprises two distinct subsets: (1) TreeVGR-RL-37K, consisting of 37K natural-image samples with target bounding-box annotations derived from TreeVGR (Wang et al., 2025a); and (2) a general reasoning subset, containing 14K samples from OpenMMReasoner (Zhang et al., 2025a) and ArxivQA (Li et al., 2024b), which lack bounding-box

*Table 11.* **Statistics of the training data.** Our training set consists of three major parts: **TreeVGR-RL**, **OpenMMReasoner**, and **ArxivQA**, totaling 51K samples.

| Dataset Collection | Subset / Source | Samples | Target Box Annotations |
|---|---|---|---|
| **TreeVGR-RL-37K** (Wang et al., 2025a) | V* (Wu & Xie, 2024) | 32K | ✓ |
| | VisDrone (Zhu et al., 2021) | 5K | ✓ |
| **OpenMMReasoner-74K** (Zhang et al., 2025a) | ViRL (Wang et al., 2025b) | 3.5K | ✗ |
| | Think-Lite-VL (Wang et al., 2025g) | 1.5K | ✗ |
| | DocVQA (Mathew et al., 2021) | 3K | ✗ |
| | MMK12 (Meng et al., 2025) | 1K | ✗ |
| | TQA (Kembhavi et al., 2017) | 1K | ✗ |
| | We-Math (Qiao et al., 2025) | 2K | ✗ |
| **ArxivQA-100K** (Li et al., 2024b) | – | 2K | ✗ |
| **Total** | | **51K** | - |

*Table 12.* **Evaluation results using different training data splits.** "37K" and "14K" denote the training data from TreeVGR-RL-37K and OpenMMReasoner, respectively.

*(a)* Natural image VQA benchmarks

| Models | Training Data | V* | HR4K | HR8K | MME-RW-Lite | POPE | RealWorldQA | CV-2D | CV-3D | Avg. |
|---|---|---|---|---|---|---|---|---|---|---|
| Qwen2.5-VL-7B | - | 78.5 | 69.0 | 65.1 | 44.5 | 86.3 | 68.1 | 75.7 | 73.6 | 70.1 |
| TreeVGR-7B | 37K | 83.8 | 77.1 | 73.1 | 54.9 | 87.3 | 67.3 | 76.6 | 77.2 | 74.7 |
| iVGR-7B (ours) | 37K | **88.0** | 77.8 | 75.0 | 55.2 | **89.2** | **68.8** | **78.5** | 80.7 | **76.7** |
| iVGR-7B (ours) | 37K + 14K | 86.4 | **78.3** | **75.5** | **55.6** | 88.9 | 68.6 | 78.4 | **81.1** | 76.6 |

*(b)* Chart understanding and multidisciplinary reasoning benchmarks

| Models | Training Data | ChartQA$_{test}$ | AI2D$_{test}$ | WeMath | MMStar | MMMU | MMK12 | Avg. |
|---|---|---|---|---|---|---|---|---|
| Qwen2.5-VL-7B | - | 86.4 | 83.6 | 35.3 | 63.9 | 54.4 | 53.6 | 62.9 |
| iVGR-7B (ours) | 37K | 85.8 | 84.6 | 36.7 | 65.6 | 53.3 | 53.3 | 63.2 |
| iVGR-7B (ours) | 37K + 14K | **88.5** | **85.0** | **41.1** | **66.3** | **55.2** | **56.3** | **65.4** |

annotations. The detailed dataset composition is presented in Table 11.

**Data Construction.** For the general reasoning subset, we apply a filtering strategy to select samples with appropriate difficulty. Using our cold-start Qwen2.5-VL-7B model, we generate 5 rollouts for each candidate query. We retain only those samples exhibiting a pass rate between 20% and 40% (i.e., 1 or 2 correct answers out of 5), ultimately selecting 12K samples from OpenMMReasoner and 2K from ArxivQA.

**Batch Sampling Strategy.** During training, we employ a mixed sampling strategy to construct each batch. Specifically, 45% of each training batch is composed of samples from the TreeVGR-RL-37K dataset. These samples are duplicated to prompt both the grounded stream and the textual stream, collectively occupying 90% of the batch. The remaining 10% is sampled from the general reasoning subset (OpenMMReasoner and ArxivQA). Since these samples lack bounding box supervision, they are trained exclusively via the textual stream without the consistency reward.

## C. Implementation Details

Our training pipeline contains two stages, *cold-start* and *RL*. The implementation details are as follows.

**Cold-Start Stage.** In this stage, we fine-tune the models on the constructed 35K dataset for one epoch using the AdamW optimizer (Loshchilov & Hutter, 2017) with Llama-Factory framework (Zheng et al., 2024). We set the global batch size to 64 and employ a cosine learning rate schedule with an initial learning rate of 1e-5.

**RL Stage.** All RL experiments are conducted using the VeRL framework (Sheng et al., 2024). We optimize the models using AdamW with a constant learning rate of 1e-6 and a KL penalty $\beta = 0.01$. For rollout generation, we set the

*Table 13.* **Performance comparison with the cold-start models.**

| Benchmarks | Qwen2.5-VL-7B | +Cold-Start | +iVGR | Qwen3-VL-8B | +Cold-Start | +iVGR | Qwen3-VL-32B | +Cold-Start | +iVGR |
|---|---|---|---|---|---|---|---|---|---|
| *Fine-grained VQA* | | | | | | | | | |
| V* | 78.5 | 80.1 | **86.4** | 82.7 | 79.6 | **90.1** | 83.8 | 84.3 | **93.2** |
| HR4K | 69.0 | 73.2 | **78.3** | 76.5 | 77.1 | **82.0** | 80.0 | 77.5 | **82.9** |
| HR8K | 65.1 | 67.5 | **75.5** | 70.4 | 71.6 | **80.1** | 78.1 | 74.9 | **82.9** |
| *General VQA* | | | | | | | | | |
| MME-RW-Lite | 44.5 | 49.5 | **55.6** | 49.0 | 51.3 | **60.7** | 52.1 | 57.1 | **61.2** |
| POPE | 86.3 | 84.4 | **88.9** | 88.1 | 82.7 | **89.4** | **89.4** | 85.8 | 88.8 |
| RealWorldQA | 68.1 | 65.4 | **68.6** | 70.5 | 69.4 | **71.0** | 79.3 | 73.1 | 76.3 |
| CV-2D | 75.7 | 73.1 | **78.4** | 78.6 | 76.5 | **80.8** | 81.2 | 78.9 | **83.9** |
| CV-3D | 73.6 | 78.8 | **81.1** | 93.5 | 88.3 | 91.0 | 92.8 | 91.1 | **93.8** |
| *Chart Understanding* | | | | | | | | | |
| ChartQA$_{test}$ | 86.4 | 86.7 | **88.5** | 83.2 | 85.8 | **87.6** | 85.0 | 87.7 | **90.4** |
| AI2D$_{test}$ | 83.6 | 82.4 | **85.0** | 80.4 | 82.2 | **85.5** | 84.5 | 87.0 | **88.7** |
| *Multidisciplinary Reasoning* | | | | | | | | | |
| WeMath | 35.3 | 34.8 | **41.1** | 49.7 | 42.9 | **55.1** | 60.0 | 55.6 | **61.6** |
| MMStar | 63.9 | 65.0 | **66.3** | 67.9 | 64.7 | **69.7** | 72.3 | 72.2 | **75.1** |
| MMMU$_{val}$ | 54.4 | 51.4 | **55.2** | 58.0 | 58.8 | **59.8** | 67.7 | 63.4 | 67.7 |
| MMK12 | 53.6 | 48.0 | **56.3** | 60.4 | 57.8 | **61.6** | 73.9 | 69.2 | **75.2** |
| **Avg.** | 67.0 | 67.2 | **71.8** | 72.1 | 70.6 | **76.0** | 77.2 | 75.6 | **80.1** |

*Table 14.* **Evaluation of the grounding capability on HR8K and TreeBench.** We report both answer accuracy (ACC) and localization quality (IoU).

| Models | HR8K | | TreeBench | |
|---|---|---|---|---|
| | ACC | IoU | ACC | IoU |
| Grounded Stream Only | 72.1 | **27.0** | 41.7 | 38.7 |
| iVGR (Grounded CoT) | 74.3 | 24.1 | 42.0 | **41.6** |
| iVGR (Textual CoT) | **75.5** | - | **44.9** | - |

sample count $N = 5$, the maximum length to 4,096, and the temperature to 1.0. Regarding the training schedule, the Qwen2.5-VL-7B and Qwen3-VL-8B models are trained for 360 steps with a global batch size of 512. For the 32B model, we accommodate hardware constraints by reducing the global batch size to 128 and increasing the training steps to 500. We employ Qwen2.5-72B-Instruct (via vLLM (Kwon et al., 2023), temperature 0.01) as the universal judge for both accuracy and consistency metrics. Crucially, in our dual-stream experiments, we maintain computational parity with baselines. We construct each batch by duplicating $B/2$ unique queries and assigning distinct prompts (textual *vs.* grounded) to each copy. This ensures that although the training strategy is "dual-stream," the total number of generated rollouts per update step remains identical to standard single-stream methods.

**Training Cost.** In our experiments, we employ 8 NVIDIA A100 GPUs for the cold-start and RL training stages, while utilizing an additional 4 NVIDIA A100 GPUs to serve the judge model (Qwen2.5-72B) via vLLM (Kwon et al., 2023). The computational overhead of the judge is minimal for two reasons: First, the judge is only required to output a final scalar score without generating long-context CoT sequences, resulting in rapid inference. Second, vLLM's support for batch inference further accelerates the consistency reward calculation. Empirically, training the iVGR-7B model takes approximately 39 hours, representing only a marginal increase over the 35 hours required when the consistency reward is excluded.

# D. Experimental Results

## D.1. Ablation Study on the Training Data Split

To ensure a fair comparison with the baseline TreeVGR (Wang et al., 2025a), we strictly align our training setup by fine-tuning the Qwen2.5-VL-7B model exclusively on the same TreeVGR-RL-37K dataset. As reported in Table 12a, under identical data conditions, our method outperforms TreeVGR-7B by 2.0% in average accuracy, validating the superior efficacy of our RL framework. Furthermore, while incorporating the additional 14K general reasoning samples yields comparable

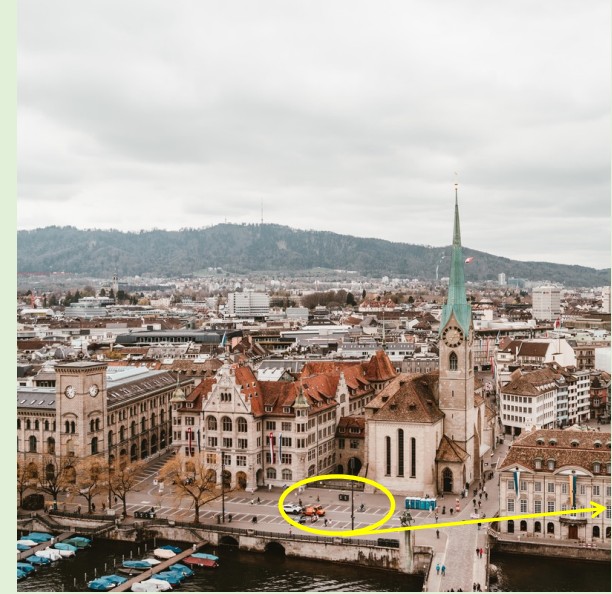
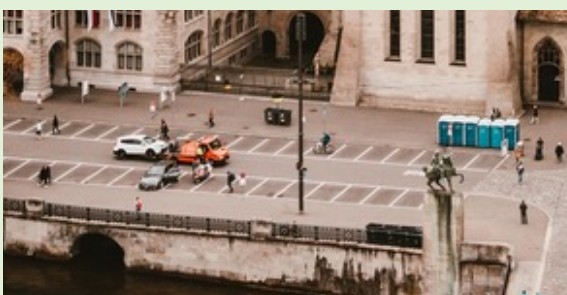

**Question**: What's the color of the trailer? **Options**: A) Black B) Orange C) White D) Blue

| **w/o Consistency reward**: \<think\>First, identify the trailer in the image. It is located near the bottom right corner, next to the orange vehicle. Then, observe its color. The trailer has a blue color with white stripes. ...\</think\> \<answer\>D\</answer\> | **w/ Consistency reward:** \<think\>First, identify the trailer in the image. It is located near the bottom right corner of the image, on the street. Then, observe its color. The trailer is clearly orange. There are no other trailers or objects that could cause confusion about the color. ...\</think\> \<answer\>B\</answer\> |
|---|---|

*Figure 5.* **Qualitative comparison between models trained with and without consistency reward.**

performance on these specific natural image VQA benchmarks, we retain this subset in our final configuration to preserve the model's robust generalization capabilities across diverse visual domains.

### D.2. Comparison with the Cold-Start Model

We report detailed performance of cold-start models via textual CoT in Table 13. Although cold-start models exhibit mixed results due to potential forgetting in general domains, our method consistently outperforms both baseline and cold-start models. Our method achieves the highest average accuracy across all model sizes (Qwen2.5-VL-7B, Qwen3-VL-8B, Qwen3-VL-32B), verifying that our RL framework enhances fine-grained VQA without compromising general reasoning capabilities.

### D.3. Ablation Study on the Grounding Capability

We study the impact of our method on the grounding capability in this section. Specifically, we evaluate the localization quality (IoU) and answer accuracy of the grounded CoT produced by iVGR, and compare it against the grounded-stream-only baseline. We additionally include TreeBench (Wang et al., 2025a) to provide a more challenging evaluation of fine-grained localization. As reported in Table 14, iVGR retains comparable IoU to the grounded-stream-only baseline (24.1 *vs.* 27.0 on HR8K; 41.6 *vs.* 38.7 on TreeBench), while consistently improving answer accuracy (+2.2% on HR8K and +0.3% on TreeBench). This indicates that jointly optimizing the policy with the textual stream does not erode the explicit grounding capability acquired from the grounded stream. Moreover, when iVGR is evaluated with textual CoT, the answer accuracy further improves to 75.5% on HR8K and 44.9% on TreeBench, consistent with our central observation that the visual localization capability is effectively internalized into the textual reasoning process.

## E. Qualitative Results

**Effect of the Consistency Reward.** We first present a qualitative comparison between iVGR and the variant trained without the consistency reward in Figure 5. The variant without the consistency reward exhibits a clear localization deviation:

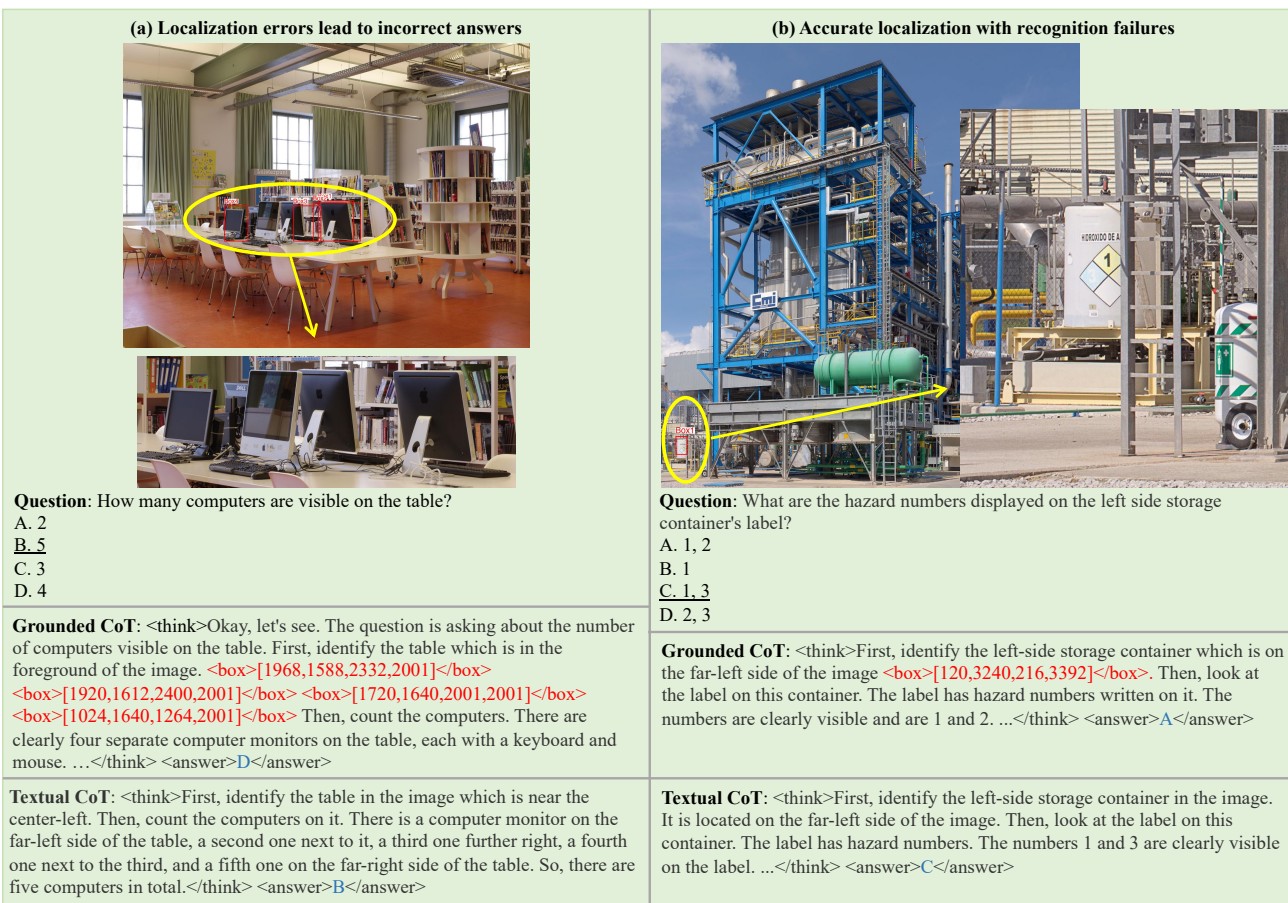

*Figure 6.* **Qualitative comparison between grounded CoT and textual CoT in our method.**

it associates the trailer with a nearby region containing a blue object, and consequently reports an incorrect color. In contrast, iVGR correctly attends to the actual trailer region and identifies its color as orange. This example illustrates how the consistency reward sharpens the textual stream's implicit localization, leading to more reliable visual descriptions in the reasoning trace.

**Grounded CoT *vs.* Textual CoT in iVGR.** We further compare the grounded CoT and the textual CoT produced by iVGR on two representative failure cases of the grounded CoT in Figure 6. The errors of the grounded CoT fall into two categories. *(1) Localization errors that propagate to incorrect answers.* In the first example, the grounded CoT predicts bounding boxes that cover only a partial subset of the computers on the table, and the final count is therefore wrong. The textual CoT, in contrast, enumerates the computers along the table without committing to explicit coordinates, and arrives at the correct answer. *(2) Accurate localization with recognition failures.* In the second example, the grounded CoT correctly localizes the left-side storage container, but misreads the hazard numbers on its label. The textual CoT, freed from emitting explicit coordinates, allocates more of its reasoning to interpreting the label content and recovers the correct numbers. These two cases together suggest that the textual CoT, by internalizing the localization capability rather than explicitly producing coordinates, is more robust to both localization noise and the distraction caused by coordinate emission during reasoning.

