# OpenReview forum: "iVGR: Internalizing Visually Grounded Reasoning for MLLMs with Reinforcement Learning"
_ICML.cc/2026/Conference — ICML 2026 regular_

### Official Review · Reviewer_SB8f · 2026-03-05

**Soundness:** 4
**Presentation:** 3
**Significance:** 2
**Originality:** 3
**Overall Recommendation:** 5
**Confidence:** 4

**Summary:**

This paper introduces iVGR, a reinforcement learning (RL) framework designed to internalize visual grounding capabilities into text-only Chain-of-Thought (CoT) reasoning in multimodal large language models. The authors first demonstrate that text-only CoT usually outperforms grounded CoT, even for models trained with grounding supervision. Motivated by this observation, they propose a dual-stream RL training strategy in which, for each input, the model generates both a grounded CoT stream and a text-only CoT stream. In addition to conventional rewards for each stream, the method introduces a consistency reward between the text-only stream and the best rollout from the grounded stream.

Through extensive experiments, the authors report improvements over prior visually grounded reasoning methods. The paper also includes ablation studies examining the contribution of individual components, the consistency reward, and training dynamics.

**Compliance With Llm Reviewing Policy:**

Affirmed.

**Final Justification:**

The new experiments in the rebuttal addressed my concerns and strengthened the paper.

**Key Questions For Authors:**

1. Could the authors clarify how gradients from the two streams are combined during training? Specifically, how are the advantages from the grounded and textual streams aggregated within GRPO? Are they weighted equally, or is there an explicit weighting mechanism?

2. Could the authors provide further evidence that grounding capability is truly internalized rather than implicitly bypassed? For example, is there an analysis showing that the textual stream maintains or reflects accurate localization behavior without explicit box outputs?

3. There appear to be discrepancies between the reported V* results and those in the TreeVGR paper: some numbers match TreeVGR exactly, while others differ. Could the authors clarify the causes of these differences?

**Limitations:**

Yes

**Strengths And Weaknesses:**

**Strengths**

1. The empirical motivation is clear and well-supported. The authors demonstrate that text-only CoT can outperform grounded CoT even for models trained with grounding supervision.
2. The experimental evaluation is comprehensive, covering multiple fine-grained and general VQA benchmarks, several open-source baselines, and different model scales.
3. The paper includes thorough ablation studies, including component analysis, threshold sweeps, judge-size comparisons, and training-dynamics analysis.
4. The reward structure and dual-stream training setup are clearly defined.
5. The idea of internalizing visual grounding into textual CoT through a dual-stream RL framework with a consistency objective is a thoughtful and practically useful extension of prior grounded reasoning approaches.

**Weaknesses**

1. The introduction is overly dense and presents detailed methodological explanations early on, which makes the paper harder to follow. Some technical details could be moved to later sections or the appendix to improve clarity.
2. There appear to be metric inconsistencies with TreeVGR (e.g., reported V* scores differ from those in the TreeVGR paper). The evaluation protocol and prompt settings should be clarified to ensure a fair comparison.
3. The experiments do not include proprietary models, which would help better contextualize the practical impact of the method.
4. Since the method reuses TreeVGR training data and reward design, while also positioning TreeVGR as a primary baseline, including results on TreeBench would strengthen the fairness and completeness of the empirical comparison.
5. The tool-assisted test-time scaling section feels orthogonal to the core contribution and could be better integrated into the main narrative or moved to supplementary material.
6. The proposed internalization primarily concerns bounding-box-level grounding. It remains unclear whether this translates to broader improvements in internal visual reasoning beyond localization-centric tasks.

---

> ### Author Rebuttal · Authors · 2026-03-31
>
> We sincerely thank the reviewer for the insightful comments.
>
> ***Weaknesses***
>
> **W1: Writing of the introduction.**
> **A1:** In the revision, we will move the technical details from the introduction to the method section to improve clarity and flow.
>
> **W2: The reported V\* score of TreeVGR.**
> **A2:** The performance discrepancy arises from differences in evaluation protocols. Following Deepeyes, TreeVGR uses a custom script for the V* benchmark. **In this custom script, ground-truth answers for all questions are option 'A'.** In contrast, VLMEvalKit preprocesses the V* data by shuffling the options, which we believe provides a more rigorous and reasonable assessment. Furthermore, recent works such as DeepEyesV2 and Thyme adopt VLMEvalKit for V* evaluation. Thus, we follow them and use VLMEvalKit for all benchmarks. Since TreeVGR utilizes VLMEvalKit for all benchmarks except V*, the performance metrics on other benchmarks for TreeVGR cited in our manuscript are taken directly from their original paper.
>
> **W3: Performance of proprietary models.**
> **A3:** We supplement the evaluation results for Gemini-3.1-Pro-Preview and GPT-5.4 as follows.
> ||V*|HR4K|HR8K|MME-Real-Lite|POPE|RealWorldQA|CV-2D|CV-3D|Avg.|
> |-|-|-|-|-|-|-|-|-|-|
> |Gemini-3.1-Pro-Preview|87.4|88.9|88.1|55.8|88.0|83.5|85.0|94.6|**83.9**|
> |GPT-5.4|88.0|87.4|80.6|63.4|87.9|83.0|82.4|91.9|83.1|
> |iVGR-Qwen3-VL-32B (ours)|93.2|82.9|82.9|61.2|88.8|76.3|83.9|93.8|82.9|
>
> **W4: Performance on the TreeBench.**
> **A4:** TreeBench is proposed by TreeVGR. ***Since the official codebase of TreeVGR is not publicly available, we were unable to exactly reproduce their setup, particularly on TreeBench.*** Instead, we implement a faithful 'Grounded Stream Only' baseline following the paper and with the help of the authors. Furthermore, our method is built based on this baseline. As shown in the table below, our model significantly improves upon this baseline across all benchmarks (e.g., boosting the TreeBench score from 41.7 to 44.9).
> Furthermore, our method outperforms the officially reported scores of TreeVGR on HR4K, HR8K and MME-Real-Lite, clearly demonstrating the effectiveness of the proposed approach. We will release our code to ensure reproducibility.
> |Models|TreeBench|HR4K|HR8K|MME-Real-Lite|
> |-|-|-|-|-|
> |TreeVGR (reported in the original paper)|**50.4**|77.1|73.1|54.9|
> |Grounded Stream Only (reproduced, our baseline)|41.7|76.1|72.1|54.1|
> |iVGR (grounded CoT)|42.0|77.1|74.3|55.3|
> |iVGR (textual CoT)|44.9|**78.3**|**75.5**|**55.6**|
>
> **W5: Writing of the tool-assisted test-time scaling section.**
> **A5:** We will move this section to the supplementary material.
>
> **W6: The generalization beyond localization-centric tasks.**
> **A6:** Please see our response to ***Weaknesses W2 of Reviewer JzB5***.
>
> ***Quesions***
>
> **Q1: Gradients from the two streams.**
> **A1:** We compute the losses for the grounded CoT and the textual CoT independently, and then directly sum them equally, without any explicit weighting mechanism.
>
> **Q2: The evidence of internalizing grounding capability.**
> **A2:** We analyze the attention scores on image tokens at the final layer across three benchmarks for our model, the 'grounded stream only' baseline, and the 'textual stream only' variant, as below. The detailed distribution comparisons are shown in [Figure Link](https://anonymous.4open.science/r/anonymize-4604/2.png).
> Experimental results demonstrate that iVGR with textual CoT and the grounded CoT exhibit similar levels of visual attention, both of which are significantly higher than the 'textual stream only' variant. The results suggest that our method enhances the textual CoT's awareness of visual evidence.
>
> |Attention to image tokens|HRBench4K|HRBench8K|MME-Real-Lite|
> |-|-|-|-|
> |Textual CoT only|0.0959|0.0960|0.1107|
> |Grounded CoT only|0.1113|0.1115|0.1258|
> |iVGR (textual CoT)|**0.1239**|**0.1235**|**0.1320**|
>
> Furthermore, we provide a concrete qualitative example below. Our model achieves accurate visual localization, whereas the variant w/o consistency reward suffers from noticeable localization deviation.
> **example:**
> [Image Link](https://anonymous.4open.science/r/anonymize-4604/4.png)
> Query: What's the color of the trailer? Options: A) Black B) Orange C) White D) Blue
> GT answer: B
> **w/o Consistency reward:** \<think>First, identify the trailer in the image. It is located near the bottom right corner, next to the orange vehicle. Then, observe its color. The trailer has a blue color with white stripes. ...\</think> \<answer>D\</answer>
> **w/ Consistency reward:** \<think>First, identify the trailer in the image. It is located near the bottom right corner of the image, on the street. Then, observe its color. The trailer is clearly orange. There are no other trailers or objects that could cause confusion about the color. ...\</think> \<answer>B\</answer>
>
> **Q3: The reported V\* score of TreeVGR.**
> **A3:** Please see our response to ***Weaknesses W2***.

---

> > ### Author Rebuttal · Reviewer_SB8f · 2026-04-03
> >
> > I thank the authors for their response and for providing the additional experiments. These clarifications address my concerns, and I will increase my score to 5.

---

> > > ### Author Response · Authors · 2026-04-07
> > >
> > > We thank the reviewer for the positive feedback and for raising the score to 5. We greatly appreciate the reviewer's recognition and encouragement, which have been very helpful in improving our work. We will carefully follow the reviewer's suggestions to further improve the manuscript.

---

### Official Review · Reviewer_7yMf · 2026-03-09

**Soundness:** 3
**Presentation:** 3
**Significance:** 3
**Originality:** 3
**Overall Recommendation:** 4
**Confidence:** 3

**Summary:**

This paper studies visually grounded reasoning in MLLMs. The authors observe that requiring explicit bounding boxes in chain-of-thought reasoning may reduce answer accuracy during inference. They argue that visual localization ability can be internalized into textual reasoning instead of being explicitly generated. Based on this, the paper proposes iVGR, a RL framework with a dual stream training strategy. One stream generates grounded reasoning with bounding boxes and receives localization rewards. The other stream generates textual reasoning and learns from the grounded stream through a consistency reward. Experiments show improvements on several fine grained visual question answering benchmarks.

**Compliance With Llm Reviewing Policy:**

Affirmed.

**Final Justification:**

Most of my concerns are addressed. I maintain my positive score (4 weak accept).

**Key Questions For Authors:**

1. The introduction claims that grounded reasoning often performs worse than textual reasoning. It would be helpful to provide more detailed error analysis. What kinds of errors appear in grounded reasoning? For example, are they caused by incorrect box predictions, noisy visual crops, or logical mistakes in the reasoning process?

2. The paper compares grounded reasoning and textual reasoning. However, it would also be interesting to see how the model performs without any reasoning step. If direct answering performs better, does this mean that reasoning itself may sometimes be unnecessary?

3. In the current design, the grounded stream generates a full chain of thought with bounding boxes. Would it be possible for the grounded branch to focus only on localization instead of generating a full reasoning process? Could a simpler grounding objective provide better supervision for the textual stream?

4. Can the authors provide examples to show the behavior of the final model? In particular, it would be helpful to compare (1) grounded reasoning and textual reasoning, (2) w/o consistency reward and w/ consistency reward  through qualitative results.

5. Does the computation of the consistency reward suffer from hallucination issues? If so, how can this problem be mitigated?

**Limitations:**

No. It would be better to provide more concrete examples, analysis of the current model’s limitations, and discussion of possible future research directions.

**Strengths And Weaknesses:**

Strengths

1. iVGR studies an interesting problem about the role of explicit visual grounding during reasoning and provides empirical evidence that textual reasoning can sometimes outperform grounded reasoning.

2. The proposed dual stream training framework better integrates visual localization ability into the reasoning process.

3. Experiments are conducted on multiple benchmarks and models, and the ablation studies are relatively comprehensive.

Weaknesses

The paper lacks qualitative visualization and intuitive analysis. Visual examples comparing grounded reasoning and textual reasoning would help readers better understand the model behavior. (More questions are in **Key Questions For Authors** )

---

> ### Author Rebuttal · Authors · 2026-03-31
>
> We sincerely thank the reviewer for the insightful and constructive comments. We respond to the **Weaknesses**, **Questions** and **Limitations** as follows:
>
> ***Weaknesses***
>
> **W1: Visual examples comparing grounded and textual reasoning.**
> **A1:** We will provide visual examples in the revision. Detailed examples and analysis can be found in our response to ***Questions Q1***.
>
> ***Questions***
>
> **Q1: What kinds of errors appear in grounded reasoning?**
> **A1:** We find that grounded CoT errors primarily fall into two categories:
> **(1) Localization errors lead to incorrect answers**.
> The model fails to ground relevant objects correctly, misleading the final reasoning.
> **example 1:**
> [Image Link](https://anonymous.4open.science/r/anonymize-4604/5.png)
> Query: How many computers are visible on the table?
> Options: A) 2
> B) 5
> C) 3
> D) 4
> GT Answer: B
> **Grounded CoT:** \<think>Okay, let's see. The question is asking about the number of computers visible on the table. First, identify the table which is in the foreground of the image. \<box>[1968,1588,2332,2001]\</box> \<box>[1920,1612,2400,2001]\</box> \<box>[1720,1640,2001,2001]\</box> \<box>[1024,1640,1264,2001]\</box> Then, count the computers. There are clearly four separate computer monitors on the table, each with a keyboard and mouse. ...\</think> \<answer>D\</answer>
> **Textual CoT:** \<think>First, identify the table in the image which is near the center-left. Then, count the computers on it. There is a computer monitor on the far-left side of the table, a second one next to it, a third one further right, a fourth one next to the third, and a fifth one on the far-right side of the table. So, there are five computers in total.\</think> \<answer>B\</answer>
>
> **(2) Accurate localization with recognition failures.**
> The model correctly grounds the target regions but misinterprets the visual semantics within those regions. An example is shown below:
> **example 2:**
> [Image Link](https://anonymous.4open.science/r/anonymize-4604/0.png)
> Query: What are the hazard numbers displayed on the left side storage container's label?
> Options: A) 1,2
> B) 1
> C) 1,3
> D) 2,3
> GT Answer: C
> **Grounded CoT:** \<think>First, identify the left-side storage container which is on the far-left side of the image \<box>[120,3240,216,3392]\</box>. Then, look at the label on this container. The label has hazard numbers written on it. The numbers are clearly visible and are 1 and 2. ...\</think>
> \<answer>A\</answer>
> **Textual CoT:** \<think>First, identify the left-side storage container in the image. It is located on the far-left side of the image. Then, look at the label on this container. The label has hazard numbers. The numbers 1 and 3 are clearly visible on the label. ...\</think> \<answer>C\</answer>
>
> **Q2: How does the model perform without any reasoning step?**
> **A2:** We evaluate models' performance without CoT as follows. Experimental results show that while CoT is not strictly necessary for the POPE hallucination benchmark, it is indispensable for the other benchmarks, especially high-resolution benchmarks like HRBench4K and HRBench8K. This is because POPE primarily evaluates object existence via binary questions, which relies more on direct visual recognition rather than reasoning.
> ||V*|HR4K|HR8K|MME-R-L|POPE|RealQA|CV-2D|CV-3D|
> |-|-|-|-|-|-|-|-|-|
> |Deepeyes Texutal CoT|81.7|74.9|73.1|53.5|89.2|69.7|77.9|80.8|
> |Deepeyes w/o CoT|80.1|70.8|70.0|49.1|88.5|67.5|76.0|80.8|
> |TreeVGR Textual CoT|84.3|76.9|74.7|54.7|88.4|69.5|77.7|79.3|
> |TreeVGR w/o CoT|82.2|69.6|68.3|53.5|88.8|65.9|72.4|76.7|
> |iVGR Textual CoT|86.4|78.3|75.5|55.6|88.9|68.6|78.4|81.1|
> |iVGR w/o CoT|86.4|72.9|69.8|52.2|88.8|68.5|76.2|78.8|
>
> **Q3: Would it be possible for the grounded branch to focus only on localization?**
> **A3:** Our proposed consistency reward relies on comparing the text descriptions of visual clues between the grounded and textual CoTs. Therefore, it cannot be calculated if the grounded stream outputs only bounding boxes. While decoupling localization and reasoning is an interesting idea, bridging pure boxes with complex textual reasoning is left to future work.
>
> **Q4: Qualitative comparisons?**
> **A4:** Please refer to ***Questions Q1*** for comparisons between grounded and textual reasoning, and see ***Questions Q2 of Reviewer SB8f*** for the example of the consistency reward.
>
> **Q5: Potential hallucination issues?**
> **A5:** Please see ***Limitations L2 of Reviewer ceu5*** for our detailed analysis and empirical results.
>
> ***Limitations***
>
> **L1: Examples, limitations, and future work.**
> **A1:** In the revision, we will follow the suggestion to add more qualitative examples and expand our limitation analysis (please see ***Limitations of Reviewer ceu5***). For future work, we plan to explore tool-augmented CoTs as reference CoTs.

---

> > ### Author Rebuttal · Reviewer_7yMf · 2026-04-03
> >
> > Thanks for addressing my concerns. I will maintain my positive score.

---

> > > ### Author Response · Authors · 2026-04-03
> > >
> > > We sincerely appreciate this reviewer's thoughtful and constructive feedback, which has been invaluable in strengthening our paper. The reviewer's encouraging evaluation and diligent efforts truly inspire us, and we are very grateful for the reviewer's dedication.

---

### Official Review · Reviewer_JzB5 · 2026-03-12

**Soundness:** 3
**Presentation:** 3
**Significance:** 3
**Originality:** 3
**Overall Recommendation:** 4
**Confidence:** 3

**Summary:**

This paper proposes iVGR, a dual-stream RL framework for visually grounded reasoning. The grounded stream is trained with format, answer, and box rewards, while the textual stream is optimized with an additional consistency reward to align with high-quality grounded rollouts. Experiments on Qwen2.5-VL and Qwen3-VL show gains on fine-grained VQA and several broader multimodal reasoning benchmarks.

**Compliance With Llm Reviewing Policy:**

Affirmed.

**Final Justification:**

The rebuttal adequately addresses my main concerns and increases my confidence in the paper, so I will raise my score accordingly.

**Key Questions For Authors:**

How robust is the consistency reward when the grounded stream has incomplete or inaccurate localization on high-resolution images?

**Limitations:**

Yes.

**Strengths And Weaknesses:**

Strengths:
The paper is motivated by an interesting observation: explicit grounded CoT can underperform plain textual CoT at inference time, even for models trained with grounded reasoning. The proposed dual-stream design is intuitive, and the empirical study is fairly comprehensive, with multiple ablations on the consistency reward, archive design, and judge setup.

Weaknesses:
The method aligns textual CoT to grounded references, but in high-resolution settings the grounded stream itself may still miss critical details, especially when grounding is represented only by bbox outputs rather than explicit crops. In such cases, the textual stream may be encouraged to follow an incorrect visual interpretation, and the paper does not provide enough analysis of this failure mode.

A second concern is attribution of the reported gains on chart and broader reasoning benchmarks. The final training setup adds extra textual reasoning data from OpenMMReasoner and ArxivQA, so it is hard to tell how much of the improvement comes from the iVGR mechanism itself versus the additional domain-relevant reasoning data.

---

> ### Author Rebuttal · Authors · 2026-03-31
>
> We sincerely thank the reviewer for the insightful and constructive comments. We respond to the **Weaknesses** and **Questions** as follows:
>
> ***Weaknesses***
>
> **W1: The impact of localization quality in grounded CoT on the consistency reward.**
> **A1:** We agree that low-quality grounded CoTs may hallucinate or miss details. To guarantee the quality of reference CoTs for our consistency reward, we filter for correct answers with a box IoU ≥ 0.3 and maintain the best historical CoTs in a rollout archive.
> We further analyze the impact of the localization quality of the grounded CoT on the consistency reward from three aspects, as follows:
> **(1) About 77\% of training questions yield high-quality reference CoTs (IoU ≥ 0.3):**
> Grounding quality naturally improves during training via the box reward of the grounded stream. Tracking the highest box IoU per question shows that only ~23% of questions lack a qualified reference CoT. For these questions, the consistency reward is strictly set to 0, preventing any misleading supervision to the textual CoT.
> |IoU Interval|[0, 0.1)|[0.1, 0.3)|[0.3, 1]|
> |-|-|-|-|
> |Ratio|11.4\%|11.3\%|77.3\%|
>
> **(2) The performance of our method is positively correlated with the grounding quality of the reference CoT:**
> To study the impact of the localization quality of the reference CoT on our consistency reward, we conduct ablation studies by selecting the reference CoT based on different IoU intervals. Specifically, we select a grounded CoT that yields a correct answer and has a box IoU in a specific interval as the reference CoT. Additionally, we utilize a rollout archive to save historically qualified reference CoTs.
> As shown in the table below, our experimental results demonstrate that the performance of our method is positively correlated with the grounding quality of the reference CoT.
> Importantly, our approach remains superior to the baseline even when restricted to low-quality reference CoTs in the [0, 0.1) interval, validating its strong robustness to the localization quality.
> When using the low-quality IoU intervals of [0, 0.1) and [0.1, 0.3), we find that the average consistency reward remains low and increases slowly during training (see [Figure Link](https://anonymous.4open.science/r/anonymize-4604/1.png)). We attribute this to the following mechanism: if a low-quality reference CoT contains hallucinations, the textual CoT is only encouraged if it generates the same hallucinated content; otherwise, the textual CoT receives a very low reward. Therefore, the consistency reward rarely encourages the textual CoT to replicate the hallucinated content from the reference CoT.
> The detailed comparison across different IoU intervals is as follows:
> |IoU Interval|V*|HR4K|HR8K|MME-Real-Lite|MMStar|Avg.|
> |-|-|-|-|-|-|-|
> |w/o consistency reward|87.4|75.3|72.4|55.0|64.7|71.0|
> |[0, 0.1)|86.9|76.5|74.0|55.0|65.3|71.5|
> |[0.1, 0.3)|86.9|77.8|74.1|55.7|64.9|71.9|
> |[0.3, 1] (ours)|86.4|78.3|75.5|55.6|66.3|**72.4**|
>
> **(3) The grounded stream in our method can be substituted with a crop-tool-augmented reasoning mechanism (e.g., Deepeyes) during training:**
> We agree that without a cropping tool, the model may miss some details in the grounded CoT. On the one hand, during training, we ensure that the selected reference CoT yields the correct answer, indicating that the CoT contains the necessary information to answer the question. On the other hand, during test time, we incorporate a cropping tool into our method to further extract details from the image. Our experiments in **Table 4 of our paper** show that our method can be further enhanced by the crop tool.
> Importantly, our consistency reward measures the alignment between the image clues described in the reference CoT and the textual CoT. Thus, this consistency reward can also be applied between a crop-tool-augmented CoT and the textual CoT. We will explore this implementation in future work.
>
> **W2: Ablation study on broader reasoning benchmarks.**
> **A2:** We use 37K natural image samples with bboxes and 14K multidisciplinary samples (details and natural image ablations in **Appendix Tables 8 & 9**). As shown below, even when trained exclusively on the 37K natural images, our method achieves slightly better results on the chart and multidisciplinary benchmarks. This validates that our method does not conflict with pure textual reasoning capabilities. As expected, adding the 14K domain-specific textual reasoning data further yields significant performance gains.
> ||Training Data|ChartQA|AI2D|WeMath|MMStar|MMMU|MMK12|Avg.|
> |-|-|-|-|-|-|-|-|-|
> |Qwen2.5-VL-7B|-|86.4|83.6|35.3|63.9|54.4|53.6|62.9|
> |iVGR-7B|37K|85.8|84.6|36.7|65.6|53.3|53.3|63.2|
> |iVGR-7B|37K+14K|88.5|85.0|41.1|66.3|55.2|56.3|**65.4**|
>
> ***Questions***
>
> **Q1: Robustness to inaccurate localization?**
> **A1:** We evaluate this robustness by controlling the localization quality of reference CoTs. Please see our response to ***Weaknesses W1*** for detailed analysis.

---

> > ### Author Rebuttal · Reviewer_JzB5 · 2026-04-03
> >
> > The rebuttal adequately addresses my main concerns and increases my confidence in the paper, so I will raise my score accordingly.

---

> > > ### Author Response · Authors · 2026-04-03
> > >
> > > Thanks for the reviewer's insightful and constructive comments, and for the tremendous effort the reviewer has put into helping us improve our paper. We are also truly encouraged by the reviewer's recognition, and we greatly appreciate the reviewer's valuable contribution.

---

### Official Review · Reviewer_ceu5 · 2026-03-13

**Soundness:** 3
**Presentation:** 3
**Significance:** 3
**Originality:** 3
**Overall Recommendation:** 5
**Confidence:** 4

**Summary:**

This paper studies visually grounded reasoning in multimodal large language models (MLLMs). Existing approaches often enforce explicit grounding during reasoning by generating bounding boxes or invoking cropping tools within chain-of-thought reasoning. However, the authors empirically show that explicit grounding at inference time does not always improve performance, and can even degrade reasoning when localization predictions are inaccurate.

To address this issue, the paper proposes a novel rollout strategy during post-training MLLMs: using high-quality grounding-based reasoning to reward textual based COT and only conduct gradient descent based on textual COT. This is a novel method for using the policy itself as reward model.

**Compliance With Llm Reviewing Policy:**

Affirmed.

**Key Questions For Authors:**

1. How large must the rollout archive be for effective training?

2. What is the rollout size for both textual and Grounding COT?

3. What are the max pixels for the evaluation? I know models' performance on some of the reported benchmarks are quite sensitive to this hyperparameter.

**Limitations:**

Limitations:

1. Dependent on bbox on annotation and tasks that requires such grounding.
2. The reliance on an external LLM introduces uncertainty.

Typos:
1. Table 1 should be ``Qwen-2.5-VL''

**Strengths And Weaknesses:**

Strengths:
1. The paper challenges a common assumption that explicit grounding during inference improves reasoning. The analysis showing that textual chain-of-thought can outperform grounded reasoning when localization is noisy is an interesting and important observation.

2. Using high-quality grounded trajectories to guide textual reasoning through consistency scoring is a novel training signal that encourages alignment between grounded and textual reasoning processes.

3. Ample ablation studies.

Weaknesses
1. The requirement for bbox annotation can be a bottleneck for scaling up the training.

2. It is unclear if the grounding capability will be affected through training as the policy updates based on textual COT. Table5 shows the G-COT performance but did not demonstrate the grounding acc itself.

---

> ### Author Rebuttal · Authors · 2026-03-31
>
> We sincerely thank the reviewer for the insightful and constructive comments. We respond to the **Weaknesses**, **Questions** and **Limitations** as follows:
>
> ***Weaknesses***
>
> **W1:The requirement for bbox annotation can be a bottleneck for scaling up the training.**
> **A1:** While our method utilizes bbox data, this limitation can be mitigated by automated pipelines:
> (1) repurposing existing detection datasets (e.g., for the counting task), and
> (2) leveraging off-the-shelf detectors for pseudo-labeling.
> These strategies significantly reduce human effort. Furthermore, our method is compatible with general textual reasoning tasks. As shown in **Appendix Table 9**, the lack of bboxes in certain tasks does not negatively impact our model's performance, demonstrating its broad applicability. We will discuss this point in the revision.
>
> **W2: It is unclear if the grounding capability will be affected through training as the policy updates based on textual CoT.**
> **A2:** The grounding capability is not negatively affected by textual CoT updates. Our method jointly optimizes the policy using both grounded and textual branches, ensuring that the grounding capability is preserved during training.
> As shown in the table below, in which we additionally experimented on TreeBench, our method yields comparable IoU and improved accuracy to the 'grounded stream only' baseline.
> This demonstrates that incorporating textual CoT for policy updates has a negligible impact on the model's grounding performance.
> |Models|HRBench8K (ACC)|HRBench8K (IoU)|TreeBench (ACC)|TreeBench (IoU)|
> |:-|:-:|:-:|:-:|:-:|
> |Grounded Stream Only|72.1|27.0|41.7|38.7|
> |iVGR (grounded CoT)|74.3|24.1|42.0|41.6|
> |iVGR (textual CoT)|75.5|-|44.9|-|
>
>
> ***Questions***
>
> **Q1:How large must the rollout archive be for effective training?**
> **A1:** In our experiments, we maintain only one grounded CoT as the reference for each question, which is sufficiently effective for training. In future work, we will explore maintaining multiple high-quality grounded CoTs and investigate how to leverage them to calculate our proposed consistency reward.
>
> **Q2: What is the rollout size for both textual and Grounding COT?**
> **A2:** In all our experiments, we set the rollout size for both textual and grounded CoTs to 5. More implementation details are provided in **Appendix Section C**.
>
> **Q3: What are the max pixels for the evaluation?**
> **A3:** Following the default evaluation setting of VLMEvalKit[1], we set the maximum resolution to 3584x3584 pixels. TreeVGR[2] also adopts this same setting.
> [1] Vlmevalkit: An open-source toolkit for evaluating large multi-modality models, ACM MM 2024
> [2] Traceable Evidence Enhanced Visual Grounded Reasoning: Evaluation and Methodology, ICLR 2026
>
> ***Limitations***
>
> **L1: Dependent on bbox annotation.**
> **A1:** We will discuss this limitation in our revision. As discussed in our response to ***Weaknesses W1***, the annotation cost can be effectively mitigated by building automated pipelines, including the use of existing detection datasets for QA synthesis and off-the-shelf models for pseudo-labeling. This significantly lowers the requirement for manual annotation while ensuring the model's scalability.
>
> **L2: The reliance on an external LLM introduces uncertainty.**
> **A2:** We agree that an external LLM introduces uncertainty, which is a challenging problem. We can improve the quality of LLM judge scores by scaling the judge model and ensembling multiple judge scores.
> **(1) scaling the judge model:**
> In **Table 7 of our paper**, we ablate the influence of different judge models with different model sizes. Experimental results demonstrate that a larger model as a judge benefits our method more. We believe that a stronger judge model can produce higher-quality scores, and further improve the performance of our method.
> **(2) ensembling multiple judge scores:**
> We can improve the quality of judge scores by ensembling multiple sampling results. Specifically, we apply Qwen-2.5-72B as the judge model and set the sampling temperature to 0.5. Then we sample 4 judge scores for each textual CoT, and average them as the consistency reward. Experimental results show that this approach can further improve the overall performance of our method, as follows:
> |   |V*|HR4K|HR8K|MME-Real-Lite|POPE|RealWorldQA|CV-2D|CV-3D|Avg.|
> |---|---|---|---|---|---|---|---|---|---|
> |single judge score |86.4|78.3|75.5|55.6|88.9|68.6|78.4|81.1|76.6|
> |ensembling 4 scores|88.5|77.4|74.4|54.7|89.1|70.5|78.1|82.2|**76.9**|
>
> **L3: Typo in Table 1**.
> **A3:** We will correct this typo in the revision.

---

### Decision · Program_Chairs · 2026-04-30

**Decision:**

Accept (regular)

**Comment:**

This paper presents iVGR, a dual-stream reinforcement learning framework that internalizes visual grounding into text-only Chain-of-Thought reasoning for multimodal LLMs. The authors identify a counterintuitive but well-supported phenomenon: explicit grounded CoT often hurts inference performance due to noisy localization predictions. To address this, the method uses high-quality grounded rollouts to supervise a textual reasoning stream via a consistency reward. The reviewers unanimously support the paper, noting the strong empirical motivation, the intuitive training design, and the thorough evaluations across multiple VQA benchmarks and ablations. The AC agrees with the positive consensus and recommends acceptance.